# Pegasus, a small extracellular peptide enhancing short-range diffusion of Wingless

Emile G. Magny[1], Ana Isabel Platero[1], Sarah A. Bishop[1,2], Jose I. Pueyo [2], Daniel Aguilar-Hidalgo [3,4] & Juan Pablo Couso [1✉]

Small Open Reading Frames (smORFs) coding for peptides of less than 100 amino-acids are an enigmatic and pervasive gene class, found in the tens of thousands in metazoan genomes. Here we reveal a short 80 amino-acid peptide (Pegasus) which enhances Wingless/Wnt1 protein short-range diffusion and signalling. During *Drosophila* wing development, Wingless has sequential functions, including late induction of proneural gene expression and wing margin development. Pegasus mutants produce wing margin defects and proneural expression loss similar to those of Wingless. Pegasus is secreted, and co-localizes and co-immunoprecipitates with Wingless, suggesting their physical interaction. Finally, measurements of fixed and in-vivo Wingless gradients support that Pegasus increases Wingless diffusion in order to enhance its signalling. Our results unveil a new element in Wingless signalling and clarify the patterning role of Wingless diffusion, while corroborating the link between small open reading frame peptides, and regulation of known proteins with membrane-related functions.

[1] Centro Andaluz de Biologia del Desarrollo, CSIC-Universidad Pablo de Olavide, Sevilla, Spain. [2] Brighton and Sussex Medical School, University of Sussex, Brighton, UK. [3] School of Biomedical Engineering, University of British Columbia, Vancouver, BC, Canada. [4] Michael Smith Laboratories, University of British Columbia, Vancouver, BC, Canada. ✉email: jpcou@upo.es

Small Open Reading Frames (smORFs) coding for peptides of less than 100 amino-acids are emerging as a fundamental and pervasive gene class, mostly uncharacterised but numbering tens of thousands in metazoan genomes[1–3]. A rich source of smORFs are so-called long-non-coding RNAs, which can be translated[4–6] producing peptides with important functions[7–10]. Another group of smORFs (called short coding sequences or sCDSs[1]) are often annotated as coding genes of mostly unknown function, but produce peptides about 80aa long with biochemical properties and subcellular localisation typical of secreted and membrane-associated peptides[1,4]. Characterised examples include antimicrobial peptides, organelle components[1,3], and cell signals[11]. Here we expand the functional relevance of sCDSs in particular, and smORFs in general, by characterising a secreted peptide with a developmental function in the *Drosophila* wing.

Fly wing development is a well-studied developmental system where several important cell signals have been characterized. Amongst these is the secreted cell-signalling protein DWnt1/ Wingless (Wg), which has sequential expression patterns and functions in wing development[12,13], culminating in the patterning of the presumptive wing margin. Reducing Wg protein secretion or transport late in development produces effects ranging from total abolition of the wing margin to partial loss of bristles and reduction of proneural gene expression[13,14]. Here, we show that the *Drosophila* short coding sequence CG17278 is translated into a secreted peptide that interacts physically with, and increases the short-range diffusion of Wg. This enhancement is essential to establish full proneural gene expression, and hence, the full wing margin pattern. We named this gene *pegasus* (*peg*) after the mythical winged horse able to carry Greek heroes over long distances.

## Results

*peg* encodes an 80aa peptide with a signal peptide (suggesting secretion or membrane localisation) and a Kazal2/Follistatin-like domain (FS-like) (Fig. 1a). FS-like domains are present in Agrin and other heparin-sulphate proteoglycans (HSPGs). HSPGs are secreted and bind to the extracellular matrix, facilitating the anchoring or diffusion of other proteins across it; thus, HSPGs influence Wnt and Hh signalling in vertebrates and flies[15]. *peg* showed wide conservation in insects, and likely conservation in uncharacterised vertebrate genes similar to SPINK extracellular protease inhibitors (Fig. 1 and S1, Table S1)[16]. However, the pattern of conserved cysteines (red) in Peg peptides differs from both SPINK protease inhibitors and HSPGs, highlighting a new protein family (Fig. 1 and S1, Tables S1 and S2). *peg* was strongly expressed in the developing fly wing but was excluded from the presumptive wing margin, a transcription pattern complementary to that of *wingless* (Fig. 1c-d).

We generated *peg*⁻ null mutants by CRISPR-Cas9, producing small deletions within the *peg* ORF (Fig. 1b). These mutants show high pupal lethality (62%) and a significant reduction in the number of chemosensory bristles at the wing margin, a characteristic phenotype of *wg* loss of function[13] (Fig. 2a, b, e, Table S1). The chemosensory precursors are determined during late larval development from cells within two proneural bands about 3–4 cells wide, induced by Wg signalling from a stripe of Wg-expressing cells along the presumptive wing margin (Fig. 2c). In *peg*⁻ null mutants, the proneural marker *senseless* (*sens*) revealed a significant reduction in proneural band width, consistent with the reduction in chemosensory bristles (Fig. 2d, S2). These phenotypes suggest a positive role for Peg in Wg signalling. We looked at the expression of Sens in *peg*⁻ clones, to assess if Peg acted non-autonomously, like secreted proteins. Indeed, we

observed no effect within small *peg*⁻ mutant clones, but in larger clones *sens* is lost in cells located more than 3–4 cell diameters away from the nearest Peg-producing cells. Inside these large mutant clones we also observe a reduced Wg distribution, while in the smaller clones with normal *sens* expression the spread of Wg appears normal (Fig. 2f, g, S2). Distalless (Dll) expression appeared unchanged within large *peg* clones (Fig. S2) suggesting that *peg* has no effect on genes activated by lower levels of Wg signalling earlier in development (see also Fig. S8).

We generated transgenic flies expressing a *UAS-PegGFP* fusion (methods). PegGFP rescues the phenotype of null *peg* without producing artefacts (Fig. 2e), indicating that PegGFP is a bona-fide replica of native Peg. We expressed PegGFP across the developing wing margin using *ptcGal4*, and detected it outside the expression domain, showing that PegGFP was secreted (Fig. 3a, S3). Importantly, when co-expressed, secreted PegGFP and Wg co-localised (Fig. S3). To exclude possible artefacts of co-expressing *peg* and *wg* in the same cells, we generated a *peg-Gal4* line *by* CRISPR-mediated homology-directed genome editing (methods) to express *pegGFP* in the native *peg* pattern (Fig. 3b). *pegGal4*-driven PegGFP rescues the chemosensory bristle phenotype (Fig. 2e), accumulates in the extracellular space (Fig. 3c–d), and co-localises with endogenous Wg along the apico-basal axis, including in cells where neither *peg* nor *wg* are expressed (Fig. 3e, f). Furthermore, Wg was retained by PegGFP in co-immunoprecipitation experiments using protein extracts from either *pegGal4 UAS-PegGFP* wing discs (Fig. 3g), or from whole larvae expressing PegGFP ubiquitously (Fig. S3). This shows that Peg can interact physically with Wg in vivo, whether directly or as part of a protein complex.

Having established that Peg is secreted, co-localizes and interacts physically with Wg, we analysed further the effect of Peg on the distribution and function of Wg. In the absence of Peg, Wg showed a narrower distribution compared to wildtype (Fig. 4a,b, and S4), consistent with the previously observed narrower *sens*-expressing domain, loss of bristles in *peg*⁻ mutants, and the lower Wg spread in large *peg*⁻ clones (Fig. 2). Reciprocally, when PegGFP was overexpressed, we observed extended Wg diffusion and *sens* expression (Fig. 4c–f, S2). Peg had similar effects on WgGFP transport and signalling when UAS-WgGFP was either co-expressed with UAS-Peg, or expressed in *peg*⁻ mutants (Fig. 4g–i). Next, we corroborated that the positive effects of Peg on Wg transport and signalling are linked, mapping to the Wg protein. Firstly, we observed neither reduction nor increase of *wg* transcription upon loss or over-expression of Peg, as observed by in-situ hybridisation and by RT-qPCR in imaginal discs (Fig. S4). Secondly, the correlated expansion of the Wg secreted domain and of its target *sens* were linked by a similar expansion of Armadillo (Arm) nuclear localisation, a known marker of Wg signalling (Fig. 4j–o, S4). Thirdly, abolition of Wg transport was epistatic over Peg function: we observed that *Wg^NRT* flies, which only express a membrane-tethered version of Wg[17], showed phenotypes of bristle and *sens* expression loss similar to *peg*⁻ and *wg*⁻ mutants; phenotypes that over-expression of Peg was not able to rescue, and loss of function of *peg* did not enhance (Figs. 5a–f, 2e).

Overall our results show that diffusible Peg: (a) interacts with and enhances Wg spread, and (b) promotes Wg signalling acting upstream or on the Wg protein itself. The simplest hypothesis is that these effects are causally linked, such that the Peg peptide facilitates Wg protein transport, resulting in expanded Wg signalling and expression of target genes. We corroborated this hypothesis by direct observation of Wg transport dynamics in vivo, performing Fluorescence Recovery after Photobleaching (FRAP). We observed a recovery of 23% of WgGFP fluorescence by 5 min in wt discs, whereas *peg*⁻ discs reached only 11% by that

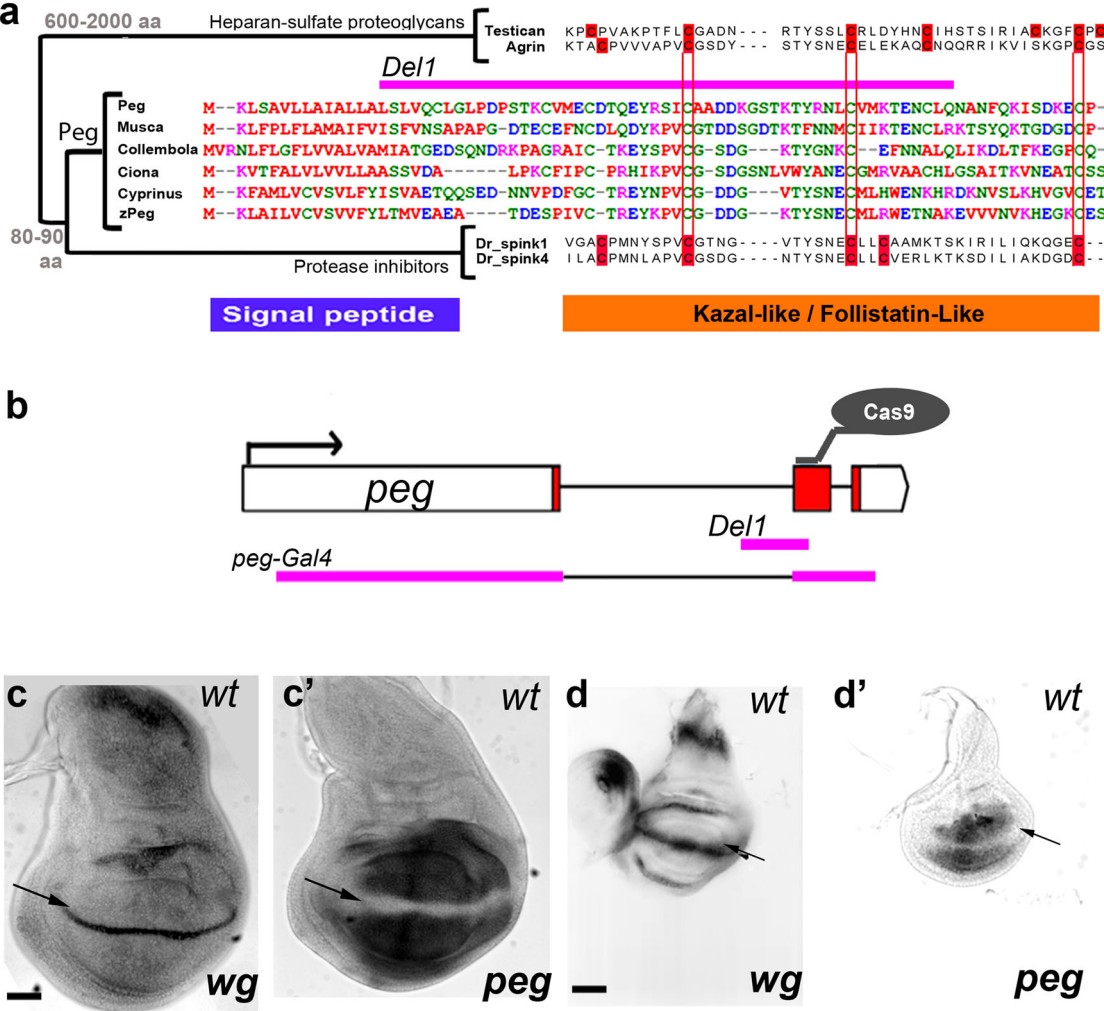

**Fig. 1 *peg* a conserved smORF is expressed in wing imaginal discs in a pattern complementary to *wg*. a** Phylogenetic tree and alignments of different proteins with FS-like domains, including Peg, the large secreted proteoglycans Agrin and Testican, and the family of small secreted Serine Protease Inhibitors of the Kazal type (Spinks). *Danio rerio* (Dr_) spinks are presented here. Size range in aa is indicated for each group of genes. Note the similarity between Peg and putative homologues in another dipteran (*Musca*), a primitive arthropod (*Collembola*), a basal chordate (*Ciona*, a tunicate), and the fishes *Cyprinus* (Carp) and *Danio rerio* (zebrafish, zPeg). The pattern of conserved cysteines (red) in each family of peptides is indicated in red. The signal peptide and FS-like domains are indicated in blue and orange, respectively. This protein structure with a signal peptide followed by a single protein domain is similar to other secreted or membrane-associated small ORF peptides, such as antimicrobial peptides[40], organelle components[39,41], and cell signals[11,42]. The *peg^Del1* allele (pink) removes a large portion of aa from the Peg sequence, including most of the Kazal domain and part of the signal peptide sequence. **b** *peg* genomic locus indicating the site targeted for CRISPR/Cas9 mutagenesis, within the *peg* ORF (red), and the genomic sequences deleted in the different alleles used in this study (pink). **c, d** In-situ hybridization for *wg* and *peg* in late (c, c´) and mid (d, d´) third instar wing imaginal discs. *peg* is expressed in a pattern complementary to *wg* (arrows, see also Butler et al.[43]). Scale bars: **c, d**: 50 µm.

time (Fig. 5g–i, Movies S1 and S2). Since Peg has a predicted domain similar to both protease inhibitors and HSPGs[15,16] (Fig. 1a), we asked whether Peg had an effect on Wg stability/degradation, or on its extracellular movement. We formalized the effective transport dynamics of Wg using the reaction diffusion model of Kicheva et al.[18] (methods; Fig. S4). We obtained a wild-type decay length for Wg ($\lambda_{wt} = 6.14 \pm 1.31$ µm) similar to Kicheva´s ($5.8 \pm 2.04$ µm), but a significant reduction in *peg*⁻ mutants ($\lambda_{peg} = 3.19 \pm 0.57$ µm), together with a significant reduction in the Wg diffusion coefficients between *peg*⁻ and wt: $D_{wt} = 0.51 \pm 0.11$ µm²/s and $D_{peg} = 0.15 \pm 0.06$ µm²/s. Further, we found no significant difference in the effective Wg degradation rates between wt and *peg*⁻ ($K_{wt} = 0.014 \pm 0.006$/s, $K_{peg} = 0.015 \pm 0.008$/s). These results imply that Wg diffuses at a slower rate in *peg*⁻ mutants, without changes to Wg degradation. We corroborated this analysis experimentally: having established above that

*peg* had no effect on Wg mRNA levels (Fig. S4), we assessed Wg protein levels using quantitative Westerns and observed no significant changes in the developing wings of *peg*⁻ mutants, nor in those over-expressing Peg (Fig. S6). Altogether our imaging, biochemical and mathematical data support a role for the Peg peptide as a secreted facilitator of Wg diffusion.

Finally, we carried out genetic interactions to explore a possible functional relationship between Peg and HSPGs, despite their different sequences and molecular sizes (Fig. 1). HSPGs have complex functions, and it has been shown that they can modulate Wg diffusion, but also that their overriding activity in this context is to bind and inactivate extracellular Wg protein in cooperation with Notum[19]. Thus, the interaction between HSPGs and Notum defines the pool of active extracellular Wg, and we would expect that Peg then interacts with this active Wg pool. We observe that the loss of wing margin bristles generated by over-expression of

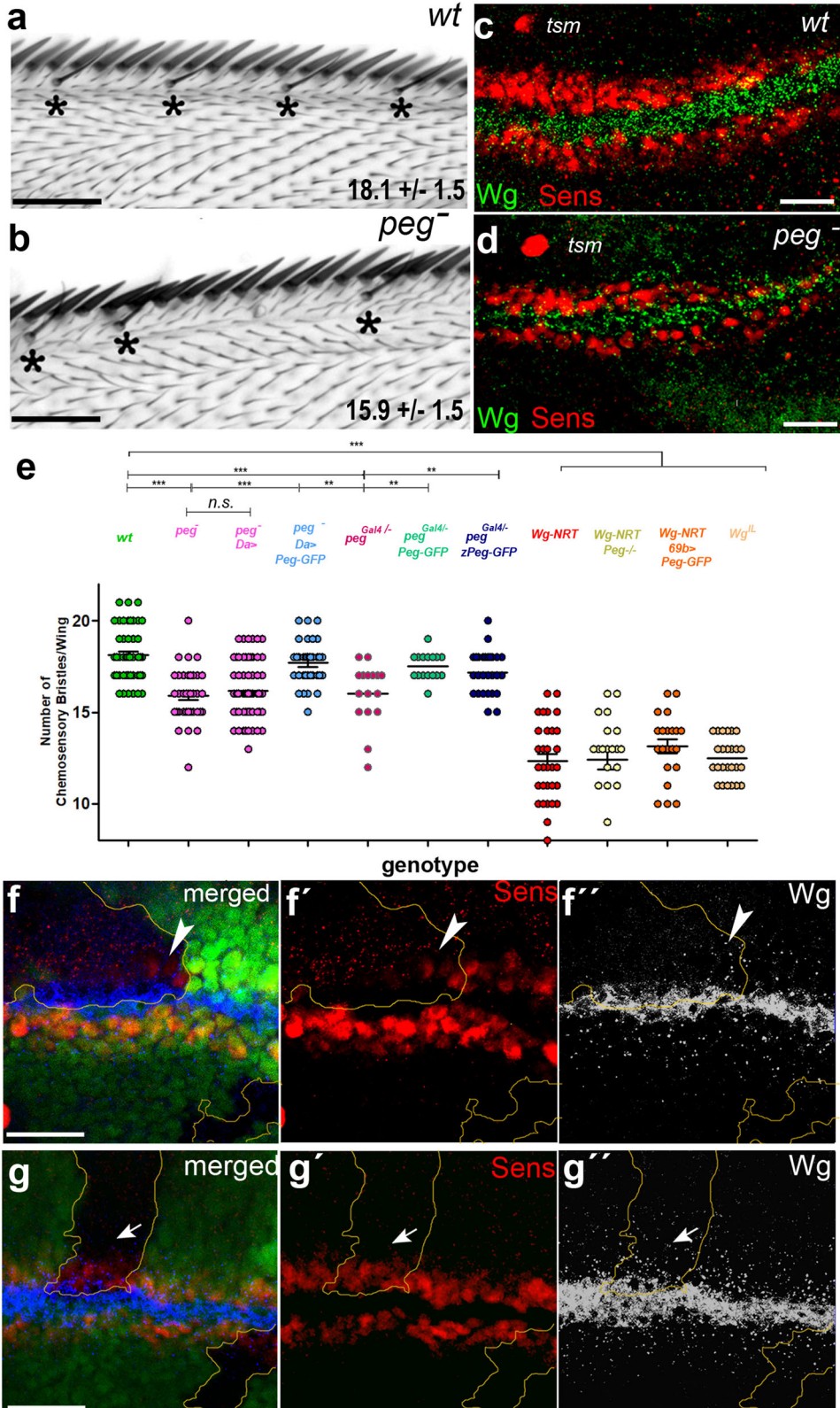

Sugarless, an enzyme involved in HSPG synthesis[20], is not rescued by over-expression of Peg (Fig. S7). This is as expected: since Peg does not generate more active Wg, extra Peg cannot rescue the reduction of active Wg caused by the *sugarless*-mediated excess of HSPGs. However, the excess of bristles created by reduction of Notum (which produces an excess of active Wg[19]) was corrected by reduction of Peg, because this excess of Wg still requires Peg to reach its targets (see Fig. S7 for details). Further, the *peg⁻* chemosensory bristle phenotype is unaffected by a reduction of the main Wg-related HSPG, Dally-like (*dlp*) and, consistently with this lack of genetic interaction, we observed no physical interaction between PegGFP and Dlp by co-immunoprecipitation (Fig. S7). Altogether our observations suggest that Peg acts on Wg separately from HSPGs.

**Fig. 2 peg has wg-like phenotypes. a, b** peg⁻ wing margins (**b**) have fewer chemosensory bristles (*) compared to wild-type (**a**), average number±SD is indicated, quantified in (**e**), **c, d.** peg⁻ wing discs (**d**) have reduced *sens* expression compared to wt (**c**; quantified in Fig. S2a-b, f). Note that the twin campaniform sensilla (*tsm*) precursor located some 3-4 cell diameters away from the *sens* dorsal stripe remains unaffected, providing a limit for Wg function (see also references[13,44], and Fig. S1). **e** Quantification of chemosensory bristles in different genetic backgrounds representing mean ± SEM. *peg* mutants, either *peg^{Del1}/peg^{Del1}*(peg⁻) or *peg^{Gal4}/peg^{Del1}* (peg^{Gal4/-}) show a significant reduction compared to wild-type. This peg⁻ phenotype is rescued by ubiquitous expression of PegGFP with *daughterless (da)-Gal4* (da > PegGFP: *w; da-Gal4, peg^{Del1} / UAS-PegGFP, peg^{Del1}*), or with *peg^{Gal4}* (Peg > PegGFP: *w; Peg^{Gal4} / UAS-PegGFP, peg^{Del1}*), which reconstitutes the endogenous *peg* expression in the wing imaginal disc. *da-Gal4* controls (*w; da-Gal4, peg^{Del1} / peg^{Del1}*) fail to rescue the number of chemosensory bristles in *peg* mutants. This peg⁻ phenotype is also rescued by expressing the zebrafish *zPeg* putative homologue with the *pegGal4* driver (Peg > zPeg-GFP). Flies homozygous for a membrane bound version of Wg (WgNRT) show a similar, albeit stronger, phenotype, which is not rescued by over-expression of PegGFP (WgNRT; 69B > PegGFP), nor worsened by its removal (WgNRT; peg^{Del1}). Wings of the *wg^{IL}* temperature-sensitive allele flies (at 17ºC) show a similar reduction of chemosensory bristles as Wg-NRT-expressing wings. (***: p < 0.001, **: p < 0.01) assessed by one-tailed t-test, See Table S1 for statistical analyses. Source data are provided as a Source Data file **f.** Large peg⁻ clones (lacking GFP, >5 cells in width, quantified in Fig. S2) show a reduction in *sens* (f,f´) and in secreted Wg (white dots) (f,f´´). Wg in expressing cells (white contours) remains normal. peg- cells neighbouring wild-type cells show near-normal expression of *sens* and extracellular Wg (Arrowheads). **g** Small clones (up to 6 cell diameters wide) show no reduction in either *sens* (g, g´) nor Wg distribution (g, g´´) (arrows). Scale bars: **a, b**: 50 μm; **c, d, f, g**: 20 μm.

Although the regulation of Wnt gradients has already a general developmental and clinical relevance[21,22], the relevance of Peg might be more direct, since we observed conservation of its sequence in chordates and vertebrates (Fig. 1a, S1). Remarkably, we corroborated the functional conservation between *Drosophila* Peg and these putative homologues in two ways. Firstly, we rescued the *peg* mutant phenotype by expressing the zebrafish homologue (*zPeg*) in the *peg* pattern (Fig. 2e). Second, we observe that zPegGFP expressed in this manner co-localized with endogenous fly Wg (Fig. S6) in the same way as found for fly PegGFP (Fig. 3).

## Discussion

Wg was characterized as a ligand diffusing over several cells[23,24], but the viability of Wg^{NRT} flies suggested that diffusion of Wg may not be actually required during development[17]. However, although Wg^{NRT} appears to reconstitute most *wg* functions, different studies have shown that Wg signalling at a distance is required in specific developmental contexts[25,26]. Here we demonstrate that short-range diffusion of Wg is required for the appropriate patterning of the wing margin and its associated proneural field, as initially reported[13]. As repeatedly shown, Wg has sequential short range signalling functions in *Drosophila* wing development, (instead of a single, long-range morphogenetic function), a crucial fact that reconciles results from several groups, including those involving Wg^{NRT}[13,14,27,28] (Fig. S8). Early in wing development (aprox. 48-72 h. after egg laying, AEL) Wg expression endows a small groups of cells with wing fate, but by the end of development two days later this instruction is inherited and indirectly felt across hundreds of descendant cells; hence, Wg perturbations at an early stage can produce either loss of the entire wing, or its duplication, without relying on long-range diffusion[12,27,29,30]. Shortly afterwards (aprox. 72-96 h AEL), transitory expression of Wg across the entire developing wing is required for wing growth[12,17,30–32], but this function does not strictly require Wg diffusion[17], since Wg is then transcribed in nearly all cells of the developing wing. Finally, during late development (96–120 h AEL), Wg is expressed in a stripe along the dorsal-ventral boundary of the wing, where its function is to pattern the wing margin[13,14]. Thus, the late patterning of the wing margin offers an immediate and direct read-out of the functional range of Wg signalling. There, complementary expression of Peg enhances Wg diffusion to reach cells located 3-4 cell diameters from the Wg source, which otherwise would be deprived of short-range Wg signal. This enhancement is essential to ensure the development of all the neural precursors and sensory organs of the wing margin. This role of the Peg small secreted peptide joins the literature showing that transport of

Wnt signals is a complex and highly modulated process, different from the elegant classical models of freely diffusing morphogens, and involves different extracellular proteins in different developmental and molecular contexts[19,33].

Our results also corroborate the growing importance of smORF-encoded peptides as cellular and developmental regulators. Given other characterised examples[1,11], their average sizes and aa composition, and their numbers[1,2], it is likely that more smORF peptides physically regulating well-known canonical proteins with membrane-related and signalling functions will be characterised.

## Methods

**Drosophila lines.** Fly stocks and crosses were cultured at 25 °C, unless otherwise stated. For time-specific activation of gene expression with the *Gal80^{ts}* system, the cultures were carried out at 18 °C and then shifted to 29 °C prior to dissection after the indicated times. The following lines were obtained from the *Drosophila* Bloomington stock centre at Indiana University (https://bdsc.indiana.edu/): *Or-R* (used as wt), *w;;69BGal4* (#1774), *w;;UASdsRed* (#6282), *w;enGal4,UAS-dsRed /CyO* (#30557), *w hsFlp;; FRT 82B tubGFP* (#5188), *da-Gal4* (#55851), *w;UAS-mCD8-FRP*(#86558) and *w; Notum*[16]/ TM3 (#4117). The UAS-*sgl* stock (M{UAS-sgl.ORF.3xHA}ZH-86Fb, #F003098) was obtained from flyORF (https://flyorf.ch/). The *w;ptcGal4-tubGal80^{ts};MKRS/TM6b* was a gift from James Castelli Gair-Hombria, the *w; UAS-wgGFP* and *w; wg^{NRT} w; dlp^{MH20}/TM6b*, and *UAS dlp-HA* stocks were gifts from Jean Paul Vincent. *rn-Gal4*[13] was previously published in St Pierre et al.[34]. *wg^{IL}* homozygous flies raised at 17 °C were obtained as in Couso et al.[13].

**Transgenic constructs.** To generate the *pegGFP* construct, a fragment containing the *CG17278* 5′ UTR and CDS sequences (devoid of stop codon) was amplified from a whole *Drosophila* embryo cDNA library and cloned into a pENTR/D-TOPO gateway vector (Invitrogen) (see Table S4 for primer sequences), this vector was then recombined with the pPWG vector (UAS-insert-C terminal GFP), obtained from the *Drosophila* Genomics Resource Centre at Indiana University (https://dgrc.bio.indiana.edu/), to obtain *UAS-pegGFP*.

The *zPegGFP* construct was generated as follows: after identifying *zPeg* (zebrafish EST *sich211-195b11.3*) as a putative vertebrate *peg* homologue, we obtained a full length zebrafish embryo cDNA library (segmentation 14-19 somites to pharyngula prim-5 stages, when *zPeg* is highly expressed according to Zfin) from the Juan Ramon Morales lab at the CABD in Seville, and used it to amplify the *sich211-195b11.3* cDNA by PCR using the zPeg Dr FW and zPeg Dr Rv 1 primers (see Table S4 for primer sequences).

We then carried out a strategy to replace the *peg* CDS by the *zPeg* CDS in the *pegGFP* construct, using the NEBuilder HiFi DNA Assembly kit (NEB), which is based on the assembly of PCR amplicons with overlapping ends. We amplified and assembled the fragments corresponding to *zPeg* CDS, flanked, upstream, by the *peg* 5′UTR, and downstream by *GFP* from the original *pegGFP* vector, using the following primers, obtained with the NEBuilder web-tool: 5′UTR_fwd, 5′UTR_rev, GFP_fwd, GFP_rev, in both cases using *pegGFP* as template. The Danio CDS fragment was amplified with the following primers: Danio CDS_fw and Danio CDS_rev using the *zPeg* cDNA amplicon as template. See Table S4 for primer sequences.

The resulting amplicons were then assembled with a XhoI-linearised pPWG vector, and sequenced.

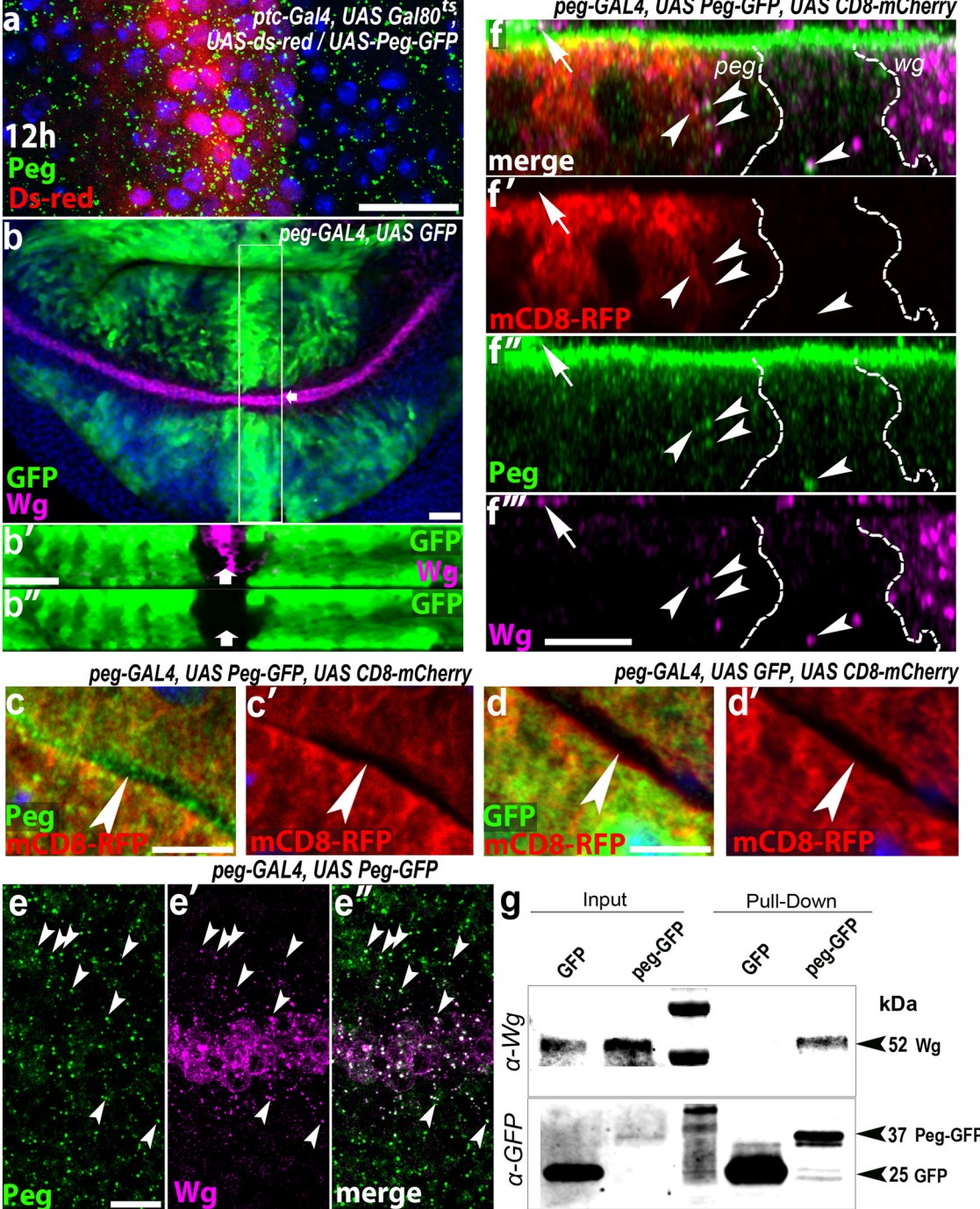

**Fig. 3 Peg is secreted and interacts with Wg. a** PegGFP (green dots) is found outside and beyond the cells expressing it (labelled with dsRed) after 12 h of induction with *ptcGal4-UASGal80ts* (methods). **b** *peg-Gal4* (methods, Fig. S9) reconstitutes the *peg* gene expression pattern in the wing pouch (see Fig.1c´,d´)), which does not overlap with *wg* gene expression as corroborated by Z-axis orthogonal reconstruction (b-b´). **c**, **d** PegGFP (green) accumulates in the extracellular space between mCD8-RFP-labelled cells (c-c´), whereas only GFP remains within the cells (red) (d-d´). **e**. PegGFP (green), expressed with *peg-Gal4*, co-localizes with Wg (magenta) in the developing wing margin cells (white dots, arrowheads). **f** An orthogonal Z-axis reconstruction shows that PegGFP (green) and endogenous Wg (magenta) colocalize basolaterally (arrowheads) within and outside of their expression domains, indicated by white dashes, and revealed by strong intracellular Wg, and *peg-Gal4* driven UAS-mCD8-RFP (peg). Notice that this co-localisation also takes place extracellularly on apical cell surfaces (arrow). **g** Pull-down with anti-GFP beads from wing discs expressing *peg-GAl4*, UAS-PegGFP yields a 50–kDa Wg-specific band revealed with α-Wg (top). GFP-only negative controls show no Wg signal. Both GFP and pegGFP were similarly bound by the pull-down beads (bottom). Source data are provided as a Source Data file. Scale bars: **a**: 20 µm; b: 30 µm; **c**, **d**, **f**: 5 µm; **e**: 10 µm.

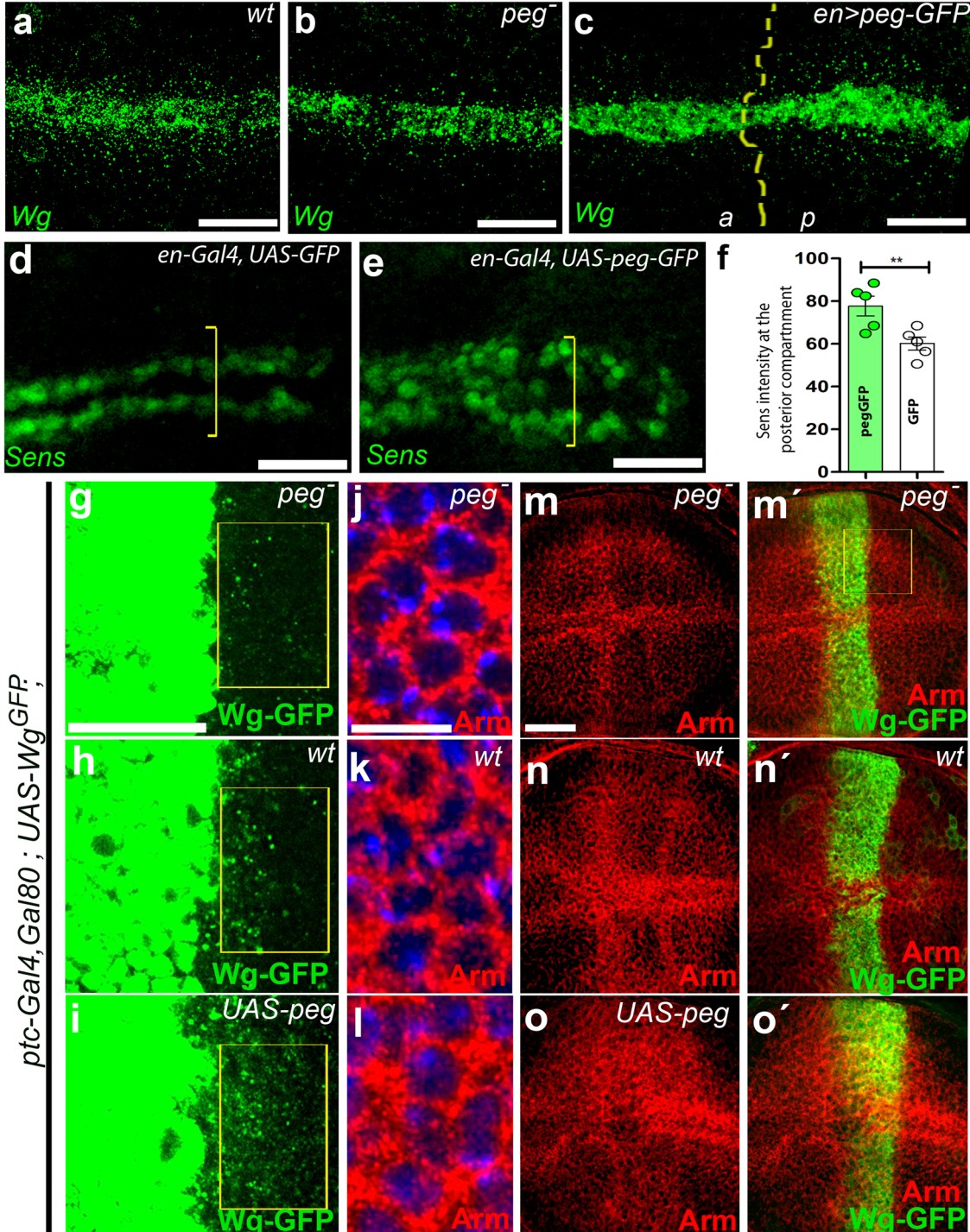

**Fig. 4 Peg enhances Wg diffusion. a–c** Wg spread is reduced in *peg⁻* wing discs (**b**) compared to wt (**a**), and is enhanced when *pegGFP* expression is driven in the posterior compartment *(p)* by *enGal4* (**c**); quantified in Fig. S4E-F. **d–f** Overexpressing *pegGFP* with *enGal4* in the posterior compartment increase *sens* expression (**e**), compared to controls expressing GFP-only (**d**); **f** Quantification from **d, e**, showing average Sens fluorescent intensity is significantly higher in posterior compartments expressing PegGFP compared to GFP controls (one-tailed t-test, $N = 10$, $P = 0.0067$, Error bars represent SEM; see also Fig. S2g-i). Source data are provided as a Source Data file. **g–i** Diffusion of WgGFP driven by *ptc-Gal4, UAS Gal80ts* (**h**) (as in Fig. 3a) is reduced in a *peg⁻* background (**g**), and enhanced when WgGFP is co-expressed with UAS-Peg (**i**) (quantified in Fig. S4g). **j–l** High magnification of the cells neighbouring WgGFP-expressing cells (yellow rectangles in **g–i**), showing Armadillo (red) and DAPI (blue). Nuclear Armadillo is reduced in *peg⁻* (**j**), and enhanced in wing discs over-expressing Peg (**l**), compared to wt *peg* background (**k**); quantification in Fig. S4H. **m–o**. Lower magnification images showing that non-membrane accumulation of Armadillo (red) in the cells near Wg-GFP (green) (**m′-o′**) is lower in the absence of Peg (**m**), and higher when Peg is overexpressed (**o**), compared to wt *peg* background (n). Scale bars: **a-e, g-i**: 20 μm; **j-l**: 10 μm; **m-o**: 40 μm.

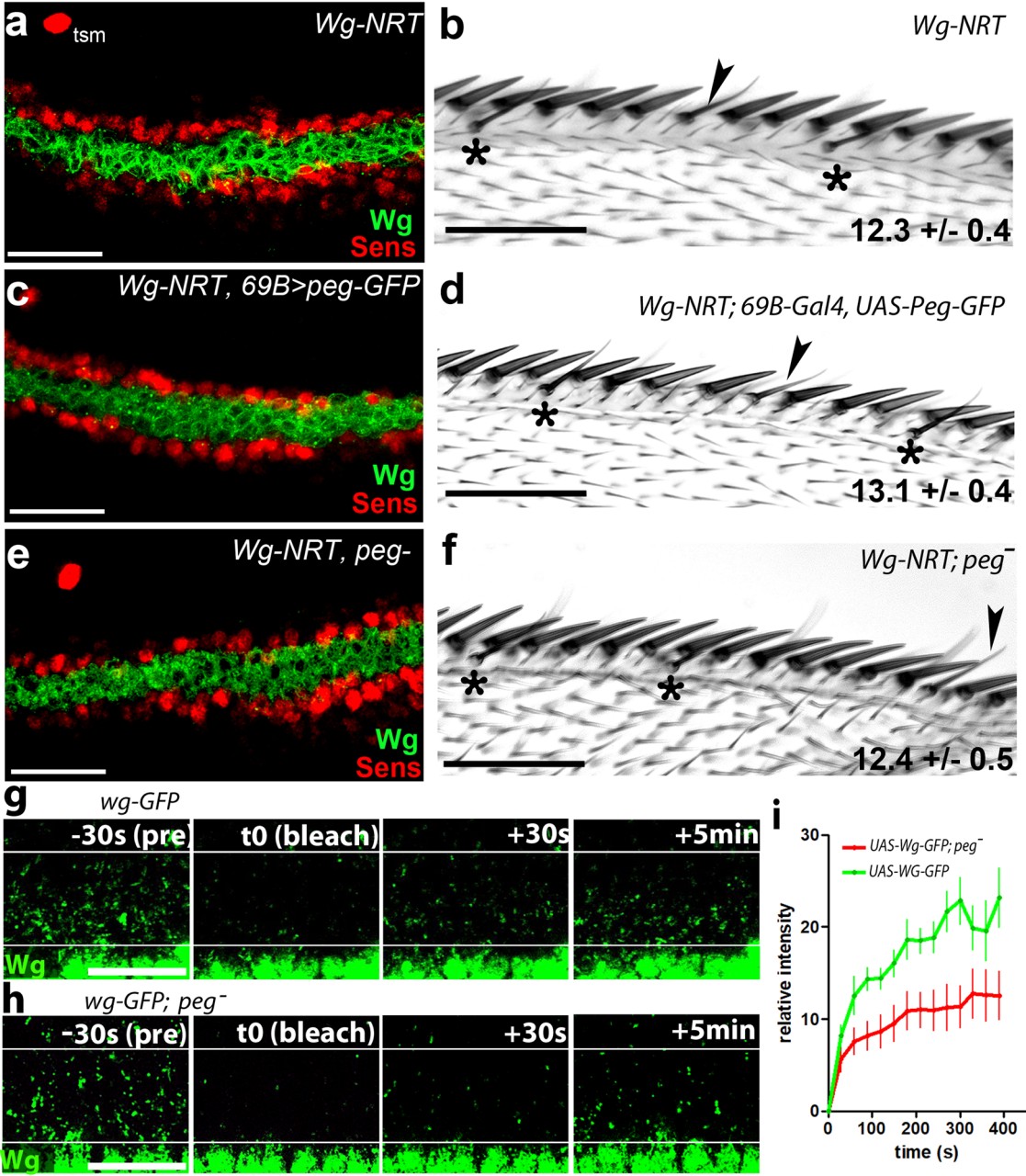

**Fig. 5 Peg acts directly on Wg protein. a** In $wg^{NRT}$ wing discs Wg (green) fails to diffuse away from the expressing cells, and Sens expression (red) is strongly reduced compared to wild type (Fig. 2c, quantifications in Fig. S2a-c, f). Note unaffected tsm. **b** $wg^{NRT}$ wings have fewer chemosensory bristles (*) (sometimes misplaced, arrowheads). **c-f** Neither the bristle phenotype, nor *sens* expression are affected by over-expression of PegGFP (c,d), or by Peg removal (**e**, **f**) (quantified in Fig. 2e). **g-h**. In-vivo FRAP imaging of WgGFP (activated by *ptc-GAL4 UAS-Gal80*$^{ts}$, 24 h before dissection) before (pre), immediately after (t0, bleach), 30 seconds, and 5 min after photo-bleaching (performed between white lines), in a wild-type (**g**) or a *peg*$^-$ (**h**) background (see also Supplementary Movies 1 and 2). **i** FRAP quantification from panels **g-h**, showing average fluorescence intensity within the bleached region, over time, relative to the intensity before bleaching (pre). Acquisition was made every 30 s. Error bars represent SEM ($n = 5$ movies per genotype). Source data are provided as a Source Data file. Scale bars: **a**, **c**, **e**, **g**, **h**: 20 μm; **b**, **d**, **f**: 50 μm.

The *UAS-pegGFP* and *zPegGFP* plasmids were sent to BESTgene for injection into embryos and generation of transgenic flies.

**Generation of CRISPR mutants**. To generate the *peg*$^-$ null mutants we cloned the following guide sequence targeting the CDS of CG17278: GTACCGATCCAT TTGCGCTGCGG into the *pU6-BbsI-chiRNA* plasmid, obtained from *addgene*, and following the available protocol (http://FlyCRISPR.molbio.wisc.edu). The following primers were used: *chi-CG17278_ORF_guide_1_Fw* and *chi-CG17278_ORF_guide_ 1_Rv* (see Table S4 for primer sequences). The *pU6-CG17278-chiRNA* plasmid was then sent for injection to BESTgene into the y[M{vas-Cas9}ZH2A w line. Surviving adults were crossed to w;;Df(3 R)BSC680, the progeny was scored for *peg*$^-$ like

phenotypes, stocks were established from putative *peg* mutants, and *peg* alleles were confirmed by PCR of the locus followed by sequencing.

**Generation of the Peg-Gal4 line (see also Fig. S9)**. The peg-Gal4 line was generated by *CRISPR*-mediated homology-directed genome editing using the strategy reported by Gratz et. al[35], co-injecting the following plasmids in a Vasa-cas9 *Drosophila* line: a pFCD4 vector carrying tandem RNA guides introduced with the primers Tan-CG17278_3'guide_Rv and Tan-CG17278_5'guide_Fw (see Table S4 for primer sequences), following the protocol described in[35]; and the *pTV[cherry]*[36] vector carrying 1 kb homology arms targeting the 5' and 3' regions of *peg* (see Fig. S9) designed to remove the whole *peg* locus while maintaining its presumptive regulatory regions. Successful *peg*$^{PTV}$ CRISPR-mediated homologous

recombinants were screened for red eyes (w+ marker within pTV[Cherry]) and sequenced. These flies expressed mCherry in some tissues but not in the imaginal discs, so we surmised that a large *peg* intronic region removed in the CRISPR-mediated homologous recombinants might carry a regulatory element required for *peg* expression in the imaginal discs. We therefore re-introduced the missing intronic region in our *peg*[pTV] lines. For this, we first carried out a Cre-Lox recombination "flip-out" by crossing the *peg*[pTV] the with a *y w; snaSco/CyO, P{Crew}DH1* line obtained from the Kyoto stock centre, to remove most of the pTV[cherry] sequence, and leaving only an ATTp site, for site-directed transgenesis. Successful flip-out events were screened by loss of the *w* marker. We then cloned the missing intronic sequence into the RIV[Gal4] vector[36], amplified using the primers intron_fragment1_fw and intron_fragment1_rv (see Table S4 for primer sequences), and EcoRI restriction. The intron-carrying RIV[Gal4] vector was then used for site-specific transgenesis in our Cre-Lox recombined *peg*[pTV] line, by co-injection with the act-phiC31-integrase vector, obtained from the DGRC (barcode 1368). Successful transformants were screened for recovery of the *w* + marker. These flies drive the expression of genes under the control of UAS in the wing imaginal discs with the same pattern of expression as *peg*; this is also a *peg* null allele since it lacks the *peg* ORF, this line is therefore called *peg*[Gal4]. Plasmid injections were made in BestGene, or at the *Drosophila* transgenesis facility at the Madrid Centre of Molecular Biology.

**Wing preparations**. For *Drosophila* adult wing preparations, the flies were collected in SH media (50% glycerol, 50% ethanol), washed in ethanol and then in dH₂O, then the wings were clipped and mounted on a slide with Hoyer´s. The slides were then placed on a hot plate at 65 °C for 3–5 h, with a weight on top of the coverslip to ensure a good flattening of the wings.

**In situ hybridization**. CG17278 ribo-probe was obtained using the *CG1728* in pENTR/D-TOPO plasmid and its T3 promoter, and the *wg* ribo-probe was obtained from a whole wg cDNA fragment in pBluescript, which was a gift from Joaquin Culi Espigul and Sol Sotillos, and it was transcribed using its T3 promoter. Digoxigenin labelling of the ribo-probes was performed with the *Roche DIG RNA labelling mix, Sigma Aldrich*, and the *Promega* T3 RNA polymerase. Wing imaginal discs were dissected in ice-cold PBS and fixed for 20 min in 4% paraformaldehyde, and a standard DIG-RNA in situ hybridization protocol, as described in Galindo et al.[7], was followed.

**Antibody stainings**. For wing imaginal disc immuno-stainings, wandering third instar L3 larvae were dissected in ice-cold PBS, cleared of digestive system and fat body, fixed for 15 min in 4% PFA, and left overnight in methanol at -20 °C. The tissues were then washed 3 times with PBS. For wingless antibody stainings, a single wash of 20 min with PBS, Tween (0.1%) was then performed, but all subsequent incubations and washes were carried out with ice cold PBS, BSA (0.2%), and ice cold PBS, respectively, maintaining detergent-free conditions in order to preserve the extracellular signal. Anti-Wg (mouse, *DSHB: 4D4-S*, used at 1:50) and anti-Arm (mouse, *DSHB: 7A1-S* used 1:50) were obtained from *The Developmental Studies Hybridoma Bank*, at the *University of Iowa* (https://dshb.biology.uiowa.edu/). Anti-Sens[37] (Guinea-Pig, used 1:3000) was a gift from Takashi Koyama. We used the following secondary antibodies: to detect Wg we used anti-mouse biotin (*Jackson, 715-065-151,* 1:200), and avidin-Cy5 (*Jackson, 016-220-084,* 1:1000) or avidin-Alexa 488 (*Jackson,* 016-540-084, 1:500), for other antigens we used anti mouse-Rhodamine, (*Jackson, 715-025-150, 1:250*), and anti guinea-pig-rhodamine, (*Jackson, 106-025-003, 1:250*). For nuclear labelling we used DAPI at a final concentration of 300 nM.

Confocal images were acquired on a Zeiss Axio Observer microscope, with an LSM 880 Airyscan module, using a 63x/1.46 oil objective. The signal from Wg, Wg-GFP, Peg-GFP, Arm, and Sens, were always detected using the Airyscan detector. The signal from DAPI and UAS-GFP, or dsRED was detected using the GAasp detectors.

**Statistics and reproducibility**. Representative micrographs from unquantified data (antibody stainings and in-situ hybridisation) presented in this manuscript were used when similar results could be observed in all imaginal discs examined, in two independent experiments. For quantified data, the number of independent biological samples analysed is indicated for each experiment.

**Activation of Peg and WgGFP expression**. In order to activate the expression of WgGFP and PegGFP at precise developmental times, *Drosophila* lines of the appropriate following genetic backgrounds: *w;ptc-Gal4-UAS-Gal80ts; peg⁻, w;ptc-Gal4-UAS-Gal80ts; UAS-peg/TM6b,* or *w;ptc-Gal4-UAS-Gal80ts,* were crossed to *w; UAS-WgGFP* or *w; UAS-WgGFP; peg⁻ /peg⁻* lines, to obtain the following genotypes: *ptc-Gal4-UAS-Gal80ts/UAS-wgGFP; peg⁻ / peg⁻, ptc-Gal4-UAS-Gal80ts/UAS-wgGFP; UAS-peg/+, ptc-Gal4-UAS-Gal80ts/UAS-wgGFP; +/+.* The progeny were reared at 18 °C for approximately 168 h and then shifted to 30 °C for 24 h. After the shift wandering L3 larvae were collected, dissected in ice cold PBS and fixed in 4% PFA. The rearing and shift times were modified as necessary for the 5 h, 8 h, 12 h, and 48 h shifts.

**Fluorescence Signal intensity measurements**. For endogenous Wg, the signal was quantified by counting the total number of Wg particles on 15 successive 2.5 micron ROIs, centred on the D/V boundary. Raw images were thresholded and the particle analyser plugin of ImageJ used to obtain the number of particles/ROI. For Induced Wg expression, the fluorescence signal profile was measured within a 20 μm² ROI in the posterior compartment, placed directly adjacent to the *ptc* domain. The Sens fluorescence signal profiles were measured within a region of interest (ROI) of 20 × 60 μm, from the centre of the d/v boundary outwards, on either side of the d/v boundary. For Arm nuclear signal quantification, we calculated the intensity of Arm signal within an ROI overlapping with DAPI signal.

**In vivo time lapse imaging**. For in-vivo time lapse imaging of WgGFP diffusion, after induction of expression, we dissected the imaginal discs from wandering L3 larvae of the appropriate genotypes, in ice cold Schneider's culture medium. The imaginal discs were then transferred to an imaging chamber similar to that used in[38]; the chamber was constructed by sticking a perforated square of double sided tape on the cover-slip of a cover-slip bottomed culture well (*MatTek Corp*), the orifice of the perforated tape was filled with culture media (Schneider's media, 2% FBS, 0.2% Penicillin-Streptomycin, 1.25 mg/mL insulin, and 2.5% methyl-cellulose as thickener to reduce drifting of the tissues during imaging). For confocal imaging, 7 section z-stacks were acquired at a rate of a whole stack every 30 seconds, using the LSM 880 Airyscan module of the Zeiss Axio Observer microscope, with a 63x/1.46 oil objective, bleaching was performed after the first stack, by 25 iterations with the 405 nm Laser line, at 80% power, in a ROI of 15 × 45 nm, placed at an average of 0.5 nm from the wgGFP source. Fluorescence intensity measurements were performed on raw image files, using Image J. In order to quantify the fluorescence recovery, we calculated the percentage of the original intensity (pre-bleach) recovered, after bleaching, for each time point, hence the following normalization for each data point: Intensity($t_X$) = (Intensity($t_x$)- Intensity ($t_{bleach}$)) *100/ intensity ($t_{pre-bleach}$).

**FRT-mediated clone generation**. In order to obtain *peg⁻* mitotic clones, we generated a *w;; peg⁻ FRT82B* recombinant line using the *w hsFlp;; FRT 82B tub-GFP* line as the source of FRT82B. These two lines were crossed, and a 1 h 37 °C heat shock was induced in the progeny 48 h after egg laying. To quantify the effect of clone size on *sens* expression we considered only clones adjacent to the *wg*-expressing wing margin cells. For this we compared *sens* fluorescent intensity within the clone, against an adjacent *sens*-expressing area of identical size within wild-type cells. The width of the clones in number of cells was determined by direct measure of the average cell-width per imaginal disc, and by dividing the average width of the clone by the average cell-width. All intensity and length measures were carried out with Image J in raw z-stacks. A total of 16 different clones were quantified, and their average intensity relative to wild-type sister cells was plotted after binning in two categories: 2–5 cells (9 clones) and 6–11 cells (7 clones).

**Co-immuno precipitation**. For co-immunoprecipitation of wing imaginal disc protein extracts, 200 discs per genotype (*w; peg*[Gal4]/*UAS-pegGFP* or *w; peg*[Gal4]/*UAS-GFP*) were homogenized in 200 μl of lysis buffer (10 mM Tris–HCl pH 7.5,150 mM NaCl,0.5 mM EDTA, 0.5% NP-40) for 30 min at 4 °C. Cellular lysate was spun at 16,000 g for 10 min at 4 °C. 300 μl of dilution buffer (10 mM Tris–HCl pH 7.5,150 mM NaCl,0.5 mM EDTA) were added to the supernatant and added to equilibrated GFP-beads (*Chromotek*), left rotating overnight at 4 °C. Beads were washed three times with washing buffer (10 mM Tris–HCl pH 7.4,150 mM NaCl,1 mM EDTA, 0.05% NP-40), and then boiled in Loading Buffer 2× (100 mM Tris–HCl pH 6.8, 4% SDS, 0.005% Bromophenol Blue, 20% Glycerol). Beads were separated with a magnet and the supernatant was loaded onto 12% Stain Free Tgx acrylamide gel (*BioRad*). Detection of specific proteins by Western Blot using a Trans Blot Turbo Transfer System (*BioRad*). Antibodies used were: rabbit anti-GFP (*Invitrogen, MA5-15256, 1:2500*); mouse anti-Wg (*DSHB, 4d4-S, 1:3000*). See next section for details of Western blots.

For larval pulldowns 60 late third instar larvae per genotype (*da-Gal4, UAS-PegGFP. da-Gal4, UAS-GFP. UAS-Dlp-HA/ +; en-Gal4 UAS-PegGFP /+. UAS-Dlp-HA/ +; en-Gal4, UAS-GFP / +*) were homogenized in 400 μl of lysis buffer (20 mM Tris–HCl pH 7.4,150 mM NaCl,1 mM EDTA, 0.5% NP-40) for 30 min at 4 °C. Cellular debris was spun at 16 000 g for 10 min at 4 °C. Supernatant was added to equilibrated GFP-beads (*Chromotek*) and left rotating overnight at 4 °C. Beads were washed three times with washing buffer (20 mM Tris–HCl pH 7.4,150 mM NaCl,1 mM EDTA), and then boiled in Laemmi Loading Buffer (*BioRad*). Beads supernatant was loaded onto 12% polyacrylamide gel and proteins were separated by SDS-PAGE (*BioRad*). Detection of specific proteins by Western Blot using a semidry blotting or tetra cell (*BioRad*). Antibodies used were: mouse anti-GFP (*Roche, 11814460001,* 1:2500); anti-Wg (DSHB, 4d4-S, 1:3000); anti-HA (*Roche, 12CA5,1:5000*).

**Quantitative Western Blots**. For each WB lane, the wing pouches of 10 wing imaginal discs were dissected and homogenised in 20 μL of LB2x Buffer (100 mM Tris–HCl pH 6.8, 4% SDS, 0.005% Bromophenol Blue, 20% Glycerol), incubated 5 min at 90 °C, and centrifuged at 16,000 × *g* for 5 min. The whole lysates were loaded on a Stain Free tgx acrylamide gel (*BioRad*). The proteins were transferred

onto a nitrocellulose membrane, with a Trans Blot Turbo Transfer System (*BioRad*) (7 min, 1.3 A, limited to 25 V). Total protein loads were quantified before and after transference, using the Image Lab suite (BioRAD). Wg was detected with anti Wg (*mouse, DSHB, 4d4-S, 1:3000*) and anti-mouse-HRP (*Donkey, Jackson, 715-035-150, 1:10000*), using the ECL select reagent (GE Healthcare, *GERPN2235*) and the Chemidoc MP imager (*BioRad*). Quantification of protein levels were performed against total protein loads for each lane.

**Quantitative real time reverse transcriptase PCR**. Total mRNA was extracted from 30 imaginal discs per genotype, using the RNeasy mini kit (*Quiagen*). For each sample, 500 ng of mRNA was used for the reverse-transcriptase reaction, using the Quantitect reverse transcriptase kit (*Quiagen*). The qPCRs were performed on a CFX connect thermocycler (*Biorad*) using the Vazyme AceQ SYBR qPCR Master Mix in 20 μL reactions, and using the Wg_fw / Wg_rv and rp49_fw / rp49_rv pairs of primers (see Table S4 for primer sequences).

**Homology searches**. For homology searches and phylogenetic analyses, we used the same methods as described in[9]. To search for sequence homologues, an initial search in ESTs deposited in NCBI (http://www.ncbi.nlm.nih.gov/) with tBLASTn with maximally relaxed parameters was carried out in dipteran species. The top 100 hits were scrutinised for belonging to a smORF of less than 100aa with start and stop codons, in the correct orientation and non-overlapping with longer ORFs. The complete smORFs passing this filter were then aligned using Clustal or MAFFT to the query and already identified orthologues of the same phylum. smORF hits showing alignment scores of at least 50 to previously identified peptides were deemed as verified. Ambiguous cases (including peptides already annotated in public databases but having unusual lengths) were tested against genomic sequences to correct sequencing or annotation errors. A consensus weighted by phylogeny was then extracted from the alignment and the process was iterated, carrying out a new tBLASTn search with the consensus sequence. When no more homologues from the same taxonomic class were obtained in a given iteration, the tBLASTn search was expanded to the next higher-order clade. Finally, phylogenetic trees for peptide sequences were generated using MAFFT and calculated according to average percent distance (see Table S2) using the unrelated 88aa small ORF membrane peptide Hemotin[39] as a potential outgroup, since it has a similar size and aa composition to the Peg peptide.

**Reaction diffusion model for Wg effective transport**. We formalize the effective transport dynamics of Wg with a simple reaction diffusion model,

$$\frac{\partial C}{\partial t} = D \frac{\partial^2 C}{\partial x^2} - KC + \nu, \tag{1}$$

where $C=C(x,t)$ is the Wg ligand concentration, which is a function of time and space, $D$ is the effective diffusion coefficient, $K$ is the effective degradation rate and $\nu = \nu(x)$ is the ligand source rate, which is spatially dependent. The steady-state solution of Eq. 1 outside of the source (target tissue) for a constant source width and and the length of the tissue in the direction perpendicular to the source being much longer than the decay-length of the gradient is given by,

$$C(x) = C_0 e^{-x/\lambda}, \tag{2}$$

where C0 is the morphogen concentration located at the target tissue right next to the source, and $\lambda$ is the decay length of the gradient, which is a function of D and K in the form,

$$\lambda = \sqrt{D/K} \tag{3}$$

**Analysis of FRAP dynamics**. Following the analysis of FRAP dynamics described in Kicheva et al.[18], we solved Eq. 1 in space and time. As initial condition at time $t = 0$, we imposed the steady state profile Eq. 2 outside of the bleached region for $x < d$ and $x > d+h$, and $C(x, t = 0) = b\, C_0\, exp(-x/\lambda)$ inside the bleached region for $d < x < d+h$, where $d$ is the distance of the ROI from the source, $h$ is the ROI width and b is the bleaching depth, that is the reduction in ligand concentration after bleaching. Then, the solution of Eq. 1 reads,

$$C(x,t) = \frac{(1-\psi)C_0}{2} e^{-x/\lambda} \left[ 1 + b + (b-1)(-A(-x,t) + e^{2x/\lambda}(-A(x,t) + A(h+x,t)) - 1 + A(h-x,t)) \right] + C_\psi(x) \tag{4}$$

Where $A(x,t) = erf\left((d + 2Dt/\lambda + x)/(2\sqrt{Dt})\right)$ with the error function $erf(z) = \frac{2}{\sqrt{\pi}} \int_0^z exp(-q^2) dq$, and $C_\psi(x)$ represents the concentration of immobile molecules that is constant in time, with $C_\psi(x) = \psi C_o e^{-x/\lambda}$ outside of the ROI, and $C_\psi(x) = b\psi C_o e^{-x/\lambda}$ inside of the ROI.

To describe the intensity recovery during FRAP, we calculate the average change in intensity within the ROI in the form $f(t) = \frac{1}{h} \int_d^{d+h} dx' C(x', t)$, using the

form of the function *f(t)* as calculated in Kicheva et al.[18].

$$f(t) = \frac{C_0}{2h} \left[ e^{-\frac{d+h}{\lambda}} \left( e^{h/\lambda} \lambda(b + (b-1)\psi + 1) - 2\lambda((b-1)\psi + 1) \right) \right.$$
$$+ (b-1)e^{-\frac{Dt+(d+h)\lambda}{\lambda^2}} \lambda(\psi - 1)\left( e^{h/\lambda} erf\left(\frac{d}{\sqrt{Dt}}\right) + e^{\frac{Dt+h\lambda}{\lambda^2}} - e^{h/\lambda} erf\left(\frac{h}{2\sqrt{Dt}}\right) + erf\left(\frac{h}{2\sqrt{Dt}}\right) \right.$$
$$+ erf\left(\frac{d+h}{\sqrt{Dt}}\right) - e^{h/\lambda} erf\left(\frac{2d+h}{2\sqrt{Dt}}\right) + e^{\frac{Dt}{\lambda^2}} erf\left(\frac{\sqrt{Dt}}{\lambda}\right) + e^{\frac{Dt+h\lambda}{\lambda^2}} erf\left(\frac{\sqrt{Dt}}{\lambda}\right) + e^{\frac{Dt}{\lambda^2}} erf\left(\frac{h\lambda - 2Dt}{2\sqrt{Dt}\lambda}\right)$$
$$- e^{\frac{Dt+h\lambda}{\lambda^2}} erf\left(\frac{2Dt + h\lambda}{2\sqrt{Dt}\lambda}\right) + e^{\frac{Dt+2d\lambda+h\lambda}{\lambda^2}}\left( -erf\left(\frac{Dt+d\lambda}{\sqrt{Dt}\lambda}\right) - e^{h/\lambda} erf\left(\frac{Dt+(d+h)\lambda}{\sqrt{Dt}\lambda}\right) \right.$$
$$\left. + \left(1 + e^{h/\lambda}\right) erf\left(\frac{2Dt + (2d+h)\lambda}{2\sqrt{Dt}\lambda}\right) \right) - erf\left(\frac{2d+h}{2\sqrt{Dt}}\right) \right] \tag{5}$$

**Determining Wg effective transport dynamics**. To find the Wg effective transport dynamics we first fit spatial concentration profiles of endogenous Wg to Eq. 2 (Fig. S5) in both *wt* and *peg*- cases. These profiles were previously normalized with respect to the intensity right next to the source $C_0$. This gives values for the decay length $\lambda$:

$\lambda_{wt}$=6.14 ± 1.31$\mu m$ (similar to the 5.8 ± 2.04 μm value in Kicheva et al.[18])
$\lambda_{peg}$=3.19 ± 0.57$\mu m$

With these values we use Eq. 5 to fit the dynamics of FRAP and extract parameters $D$, $\psi$ and $b$. From the effective diffusion coefficient for both cases:
$D_{wt}$=0.51 ± 0.11 $\mu m^2/s$
$D_{peg}$=0.15 ± 0.06 $\mu m^2/s$ and Eq. 3, we find values for the effective degradation rate:
$K_{wt}$=0.014 ± 0.006 $s^{-1}$
$K_{peg}$=0.015 ± 0.008 $s^{-1}$
Fitted values for the immobile fraction and bleaching depth read as $\psi_{wt} = 0.35 \pm 0.20$, $\psi_{peg} = 0.22 \pm 0.10$, $b_{wt} = 0.27 \pm 0.11$, $b_{peg} = 0.37 \pm 0.15$.

The fits and errors were calculated optimizing parameter search using Nonlinearmodelfit function in software Mathematica.

**Reporting summary**. Further information on research design is available in the Nature Research Reporting Summary linked to this article.

## Data availability
The authors declare no restrictions on the availability of data or biological materials (fly lines and plasmids) upon request to the corresponding author. All data-sets used for quantifications in this study are provided in the Source data file accompanying this manuscript. Pre-print Photoshop files with original high resolution images can be accessed through the Open Science Framework Repository: https://osf.io/9zt8h/?view_only = 9137cf57b96e4081a24c58c11be555f7. No custom code/software was used for this work, the set-up of the Mathematica based analysis is available upon request from Daniel Aguilar Hidalgo. Source data are provided with this paper.

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

## Acknowledgements

We acknowledge the support of Laura Tomas and Alejandro Moscoso at the CABD proteomics platform, and Katherina Garcia at the CABD Advanced Light Microscopy and Imaging Facility. We are grateful to Ana Hervas for excellent technical help, Pedro Patraquim for discussions, and Thomas Klein and Fernando Casares for comments. We thank Jean Paul Vincent for providing the Wg-NRT and UAS-Wg-GFP lines, and to Takashi Koyama for anti-Sens antibody. **Funding:** This work was funded by grants from the British BBSRC (ref. BB/N001753/1) and the Spanish MINECO (refs. BFU2016-077793-P, MDM-2016-0687) and MCI (PID2019-106227GB-I00).

## Author contributions

Emile G Magny contributed with the conceptualisation, methodology, formal analysis, investigation, original draft preparation, and visualization; Ana I Platero, Jose I Pueyo, and Sarah A Bishop contributed with the conceptualisation, methodology, investigation, and revision of the manuscript; Daniel Aguilar-Hidalgo contributed with the methodology, formal analysis and preparation and revision of the manuscript; Juan P Couso contributed with the conceptualisation, methodology, original draft preparation, and with the supervision, project administration and funding acquisition necessary to produce this work. The authors declare no competing interests, nor any restriction on data availability upon request.

## Competing interests

The authors declare no competing interests.
