## [Peer Review File · Nature Communications]

REVIEWER COMMENTS

Reviewer #2 (Remarks to the Author):

The authors describe Pegasus (Peg), a small open reading frame-encoded 80-amino acid polypeptide homologous to heparin sulfate proteoglycans (HSPG) that is required for correct short-range diffusion of Wingless (Wg). Extensive characterization of peg expression during development, pupal lethality and defective wing margin development in peg- mutants, and reduction in sens expression are provided. Using rigorous control experiments and quantitative analysis of protein diffusion, the authors demonstrate that Peg positively regulates Wg signaling through direct interaction and enhancement of diffusion rate, rather than altered Wg expression or stability. The study is significant in that it reveals a totally novel secreted factor that is required for Wg signaling, and also in that it contributes to the challenging space of identifying secreted, previously unannotated small proteins with important biological functions. The quality of the data presented are overall excellent and I have

only minor questions that may not require additional data to address.

1. It is clear that Peg and Wg interact, and that a major phenotypic consequence of peg disruption is incorrect Wg diffusion and signaling during development. However, it seems likely that Peg does not specifically interact with Wg, given the homology of Peg to HSPGs and its HSPG-like function in controlling protein diffusion. HSPGs are generally thought of as cell surface-associated proteins that display glycosaminoglycans (GAGs) for interaction with many extracellular factors (though I won't use the word "nonspecific" because I don't think it's accurate). Is it possible that Peg interacts with and affects the diffusion of multiple secreted proteins, not just Wg?

2. Related to the points above, the Peg-GFP fusion protein appears to migrate at 50 kD (Fig. 2D). I would expect a molecular weight of $27 + 8 \text{ kD} = 35 \text{ kD}$. Other microproteins exhibiting anomalous migration have been previously reported, but they are either intrinsically disordered or highly positively charged. It seems possible that the anomalously high apparent molecular weight for Peg-GFP could be due to post-translational modification with glycans, supporting its identification as an HSPG. It also seems possible that the Peg-Wg interaction could be mediated by the GAG moiety, if it exists. Especially given the paucity of high quality (and functionally characterized) identifications of post-translational modifications on small open reading frame products, this could be an interesting and field-advancing point.

3. Finally, it is debatable whether a molecule of 80 amino acids can be accurately described as a "peptide" as in the manuscript, especially if it may adopt a three-dimensional folded structure.

These are minor details given the clear phenotype, but important at the molecular level.

Reviewer #4 (Remarks to the Author):

In the article "Pegasus, a small extracellular peptide enhancing the short-range diffusion of Wingless" Couso and co-workers claim that Pegasus is a short 80 aa peptide that extends the signalling range of Wg in the *Drosophila* wing imaginal disc. The authors show that Peg can also regulate signalling of Wg by analysing target gene expression and wing morphology.

The experiments are well done, and they provide convincing evidence for a role for an association between Pegasus and Wg that is important for *Drosophila* wing development. The authors have been thorough in demonstrating that Wg and Peg co-localise, co-IP, and that Wg can enhance the distance that Wg diffuses from the cell. For the most part, the data clearly represent what is asserted in the results text, except for figure 1H, which I had to zoom in extensively to detect a reduction in secreted Wg.

I also think I would like more of an understanding from this paper about potential interactions between Peg and HSPG. The authors show that Pegasus is related structurally to HSPGs, which have been shown recently to have a role in promoting secreted Wnt signalling. It has been shown that glypicans are structurally similar to the CRD domain of Fzd receptors and thus allow long-range propagation of Wg. A similar analysis of Peg in its relation to the palmitoylation site of Wg would be crucial. It would be further interesting to show if overexpression of the two *Drosophila* glypicans can substitute for the loss of Peg.

Furthermore, given that the authors could not find evidence of a mammalian orthologue, their assertion that this is clinically relevant is somewhat unfounded. The authors claim that Peg could function comparably to Spink in vertebrates. However, overexpression of Spink5 has been shown to inhibit rather than promote β -catenin Wnt signalling in oesophageal cancer (Wang, 2019), and mutations in Spink1 are associated with pancreatic cancer and a reduction in canonical Wnt signalling (Shimosegawa, 2009).

Finally, it has been shown that tethered Wg can rescue the Wg mutant phenotype to a great extent - especially concerning the phenotype of the wing. Tethered Wg leads only to an overall reduction of 11% of the wing size without an apparent patterning phenotype. The authors could analyse the interaction of tethered Wg with Peg and need to discuss their findings in the light of this publication.

Specific points:

It is unclear to me if Peg could act as a permissive factor allowing Wg signal transduction by activation of the formation/signalling of Wnt ligand-receptor complex. The authors may show if Lrp6 signalosome formation is altered at the plasma membrane of Peg ^{-/-} cells.

The impact of this paper could be further improved by determining if Wg can bind to other Drosophila Wnt ligands, in particular for example Wnt4, which functions indirectly through PTK7 in opposition to canonical Wnt to regulate polarity, cell migration and invasion in Drosophila (Peradziryi, 2011). This could explain the role of the related Spink proteins in mammals. If OE of Drosophila peg can also function to inhibit (as well as enhance) canonical Wnt signalling in particular contexts, this may shed light on the mechanisms of Spink proteins in general and cancer in particular.

In-situ hybridisation cannot be used as a tool to quantify mRNA expression but determine localisation. (Fig. S3AB)

Reviewer #5 (Remarks to the Author):

The authors identified a novel small Open Reading Frame (smORF) peptide named Pegasus (Peg) that apparently augments the short-range diffusion of Wingless (Wg) in *Drosophila melanogaster*. Loss of Peg phenocopies a mild Wg loss of function mutant: exhibiting a reduction in chemosensory bristles in the adult wing and narrowed proneural gene expression (senseless) in the larval wing disc. Wg diffusion seems to be impeded in the large peg^{-/-} clones and in Peg mutant discs, suggesting that Peg positively promotes Wg distribution. Over-expression studies support this theory, suggesting that UAS-peg-GFP binds directly to Wg and increases Wg dispersion. The authors propose that Peg is a novel contributor to Wg signaling in the fly wing.

Major Concerns

The presented data are not sufficiently rigorous to support the conclusions made.

The Peg CRISPR mutant was not rigorously tested and only one mutant allele was used. RNAi lines and other mutant alleles that KO or KD Peg should have been used to corroborate the mutant Peg

phenotype. To test if *peg*^{-/-} is a true null, in situ hybridization experiments showing the lack of *peg* mRNA expression in a *peg*^{-/-} background should have been done like they were done in wild type discs (Figure 1C-G).

To study the over-expression of Peg, the authors used UAS-*peg*-GFP. Over-expression studies should have also been performed with UAS-*peg* (absence of tag) to ensure that tagging did not affect the function. Related to this, if Peg promotes Wg diffusion then overexpression would be expected to lead to a local reduction of Wg levels. The data in Fig S3 suggests that UAS-*peg*-GFP only leads to a mild overexpression, compared to endogenous levels.

Co-localization and expression of studies of Peg were mostly done with the UAS/Gal4 system. This is generally problematic because both Wg and Peg and were over-expressed with the same driver and the apparent co-localization could be merely co-incidental. The two secreted proteins found in same endosomes, for example. It is certainly not proof of interaction. Critically, Tween (0.1%) should not have been used during the antibody staining because this permeabilizes the membrane and it is unclear if they are really analyzing only secreted, extracellular Wg/Peg. The standard method in the field is described in Strigini and Cohen (2000).

The IP experiment in Figure 2D is missing critical controls, so it is not certain whether Wg/Peg interact with one another. Given the pivotal nature of the proposed interaction more work should be done to validate this. For example, the levels of the pulled down GFP proteins are not shown. This is essential. It would also be more convincing if the reverse experiment was done and Peg was pulled down with an anti-Wg antibody. If this is problematic, then Wg-GFP could be used and an alternatively tagged Peg. Also it is not clear if extracellular Peg and Wg interact, the pulldown was done with whole larvae lysate. This would be easy to test in cell lines. The phenotype suggests the Wg, Peg interaction is specific, it would be more convincing to test binding to another secreted protein (Grk, Dpp, Hh).

The authors suggest that Peg promotes Wg diffusion because it has an FS-like domain which are present in heparin-sulphate proteoglycans (HSPGs) that bind to the extracellular matrix and facilitate diffusion. Related to the above comments on – other extracellular factors that interact with HSPGS should be tested.

The conservation analysis is weak (Figure S1). The only conclusion that it convincing is that Peg probably has a Kazal/Follistatin-like domain. A better alignment is needed. For the work to have a significant impact on the Wnt field conservation outside of *Drosophila* would be important.

Lastly, the authors posit that only short range signalling is affected, yet the expression of long range (lower threshold) targets such as Dll are not tested. They should be.

The concept of secreted proteins modulating Wg/Wnt diffusion is not novel.

Other studies have identified similar factor which affect Wg diffusion. For example, the Nusse lab have identified Swim, a Wg binding protein, that facilitates long-range diffusion of Wg by maintaining the activity and solubility of Wg (Mulligan et al., 2012). This paper was not cited. More crucially the authors should test how Swim and Peg functionally interact. The mild phenotype of the Peg mutant suggests it is not a key factor, perhaps it is affected by Swim. For a journal with the scope and impact of Nature Communications, such considerations are de rigueur.

Minor concerns

1) Figures are poorly presented

a. Figure 2D poor image quality

b. Figure 3 F,H graphs have different scales; labeling; fonts/typos

c. Figure S3 B', stretched graphs UAS-CG17278, (Is this UAS-peg-GFP) or is it UAS-peg?

2) Methods are limited

a. The full genetic background of each genotype is not written out which prevents a critical judgement of the experiments.

b. Authors should show senseless expression in Da-Gal4 UAS-GFP and 69B-Gal4 UAS-GFP flies because these Gal4 drivers are used interchangeably and they may independently act on sens expression. Da is known to affect Sens.

c. Wg-IL and Wg-NRT have similar defects but peg alterations were only tested in the Wg-NRT background, for consistency this should have been repeated with Wg-IL (Figure S1). The number of chemosensory bristles should also be shown.

3) Conflicting/unexplained phenotypes

a. It isn't explained why is there a complete loss of sens in large clones, but only narrowed presence in mutant discs (Figure 1H,I). Indeed given the expression pattern of Peg in the pouch it is not clear why lost and not reduced. It is highly expressed in both the dorsal and ventral compartments and based on the experiments in Figure 2 be able to reach the clone where Wg is present.

b. sens expression is reduced in the control en>GFP, but this was not addressed by the authors. Thus the increased of sens expression with en>UAS-peg-GFP is questionable.

Reviewer #3 (Remarks to the Author):

Referee Report for “Pegasus, a small extracellular peptide enhancing the short-range diffusion of Wingless” by E. Magny et al

The authors show that the peptide Pegasus (*peg*) enhances Wingless/Wnt1 protein short-range diffusion and signalling during *Drosophila* wing development. They report creating *peg*' null mutants and conducting experiments to compute effective diffusion coefficients and decay lengths for Wg in FRAP experiments, where the spread of Wg was measured after photo-bleaching.

I was asked to evaluate specifically the authors use of a reaction diffusion model for Wnt/Wingless morphogen gradients.

The authors use the well-established model of Kicheva (2007), which is properly cited throughout the manuscript. Methodologically, this is very appropriate. (I do have some fairly minor comments, see below.) Overall, I found the manuscript to be well written, scientifically sound and its result novel and interesting.

Detailed comments:

- (i) page 25: " C_0 is the amplitude (intensity) of the gradient at the target tissue right next to the source" This is not quite correct. In (2), C_0 is the concentration of the morphogen itself (measured via the GFP intensity in the experiment), not of the gradient. In fact, an important aspect of Kichova et al.'s method is that the morphogen flux (=the magnitude of the gradient of C_0) at the source $x = 0$ is constant and given by C_0/λ . The authors should correct this sentence.
- (ii) p. 27 "[W]e ... fit spatial concentration profiles of endogenous Wg to Eq. 2": How was the fitting done? How were the reported error bounds determined? And how was the goodness of fit evaluated?

- (iii) I found it interesting that while the Wg degradation rates of the wild type and *peg'* are essentially the same, both the immobile fractions and the bleaching depth (=reduction in ligand concentration after bleaching) are quite distinct. Are their differences statistically significant? (The confidence intervals overlap, but that doesn't mean they're not.) And what may be the reason for this differences?
- (iv) p. 26 The "bleaching depth" should be defined to make the manuscript more self-contained. (One may think first that this is a length, but of course it's not.)
- (v) p. 25, eq. (1) I found the inclusion of the production rate ν slightly misleading since it's never given explicitly and in fact ultimately, the only way that the production of morphogens enters the model is via the assumed flux at the source boundary. (Kichova et al. use a Dirac delta distribution $\delta(x)$.) Also (slightly nit-picky) I found the different sizes of the "C" in eq. (1) to be very confusing. It looks like the "C"s in $\frac{\partial C}{\partial t}$ and $\frac{\partial^2 C}{\partial x^2}$ are different than the one in $-KC$.
- (vi) p. 26 eq. (4): The explicit solution is taken from Kichova et al. (2007) and since it's not all obvious, it would be good to indicate this in the sentence directly preceding it.
- (vii) p. 26 eq. (5) Giving the explicit form of $f(t)$ is not necessary and just bloats the manuscript.

REVIEWER COMMENTS

We would like to address first a misconception, which we have tried to correct here with more conservation data. The Peg peptides contain a stretch that has similarities to Kazal2 FS-like domains, a superfamily of domains with a very loose consensus sequence based on a pattern of repeated Cysteins with structural function. There are some 500 protein families with this type of domain, and the molecular function of such domains varies widely. In the case of Peg, the closest similarities are first, to the Spink protease inhibitors, and second, to HSPGs. These similarities can't be construed to classify Peg as an HSPG or a Spink homologue. The Peg peptides have a unique pattern of Cys that is different from both HSPGs and Spinks. In fact, our new data confirms that Pegasus belongs to a previously unidentified, widely-conserved family of peptides, present in basal chordates and arthropods and hence at least 500 million years old. Rather, the putative Kazal-FS domain in Peg offers us avenues to explore the molecular function of Peg.

Regarding a similarity with Spinks, these are a vertebrate-specific family with protease-inhibitory activity, which we show Peg not to have: Peg does not affect Wg protein levels (as revealed by quantitative Westerns, Fig. S6), nor does it change the Wg degradation rate (as revealed by Wg in vivo FRAP experiments, Figs. 5 and S6). Since Peg does not work as an anti-degradation factor, there is no reason to suppose a primary and direct functional or molecular relationship between Peg and Spinks, just as no such direct and primary functional relationship between Spinks and other Wnts has been shown to our knowledge.

Regarding HSPGs, firstly these are not altogether molecularly similar to Peg: HSPGs are large proteins with multiple Kazal domains and linked to the cell surface or the extra cellular matrix, whereas Pegs are diffusible peptides with a single putative Kazal domain. Second, our pull-down Western controls (Fig. 3) show that, unlike HSPGs, Peg is not post-translationally modified (Peg-GFP shows its expected size). Third, HSPGs directly affect both Wg diffusion and signalling, whereas Peg acts on Wg diffusion, and only affects Wg signalling indirectly, as indicated by the experiments where WgNRT (which can signal but can't diffuse) is totally epistatic over Peg (Fig. 5), and thus can't be corrected by Peg, nor itself corrects loss of Peg.

However, although we clarify that Peg is not an HSPG, we understand that the referees might still be concerned by the possibility that Peg could bind to or act primarily on an HSPG, so that the physical and functional interaction between Peg and Wg that we show would be indirect and actually mediated by HSPGs. In other words, that Peg would interact with HSPGs that would interact with Wg. In this version of the manuscript, we add results indicating that this is not the case, but rather that Peg works formally downstream of HSPGs and Notum (Fig. S7 and main text).

HSPGs have a complex function, enhancing Wg spread while also acting as "gradient sinks", cooperating with Notum extracellularly to inactivate Wg (by removing an essential palmitoleate moiety from Wg). Crucially, this inhibitory activity overrides their effect on Wg spreading (Kakugawa et al. 2015, Nature). Thus, an

increase in HSPGs leads to a reduction of active Wg, and hence to a loss of wing margin sensory organs (Avanesov et al. 2012, PLoS Genetics). This is what we observe when over-expressing *sugarless (sgl)*, an enzyme required for HSPG synthesis (Hacker et al. Development (1997), and this phenotype is not rescued by over-expressing Peg, suggesting that Peg cannot increase the pool of active Wg.

Reciprocally, we show that the ectopic chemosensory bristle phenotype caused by reduction of Notum (Kakugawa et al. 2015, Nature), is rescued by reduction of *peg*, in a double heterozygous background. In this case, reduction of Peg reduces the effectiveness of the excessive active Wg produced by Notum/+, because this excessive Wg still requires Peg to reach its targets. We further show that Peg does not interact at neither the genetic nor physical levels with the well-characterised HSPG Dally-Like, the main HSPG interactor of Wg in this process. Altogether, these results suggest that Peg acts upon the “signalling capable” pool of Wg already determined by HSPGs and Notum, .i.e. HSPGs and Peg functions converge on Wg, but act separately. We have included a figure (Fig. S7) to illustrate and explain these interactions, which we agree help to define more precisely the function of Peg.

Reviewer #2 (Remarks to the Author):

The authors describe Pegasus (Peg), a small open reading frame-encoded 80-amino acid polypeptide homologous to heparin sulfate proteoglycans (HSPG) that is required for correct short-range diffusion of Wingless (Wg). Extensive characterization of *peg* expression during development, pupal lethality and defective wing margin development in *peg*- mutants, and reduction in *sens* expression are provided. Using rigorous control experiments and quantitative analysis of protein diffusion, the authors demonstrate that Peg positively regulates Wg signaling through direct interaction and enhancement of diffusion rate, rather than altered Wg expression or stability. The study is significant in that it reveals a totally novel secreted factor that is required for Wg signaling, and also in that it contributes to the challenging space of identifying secreted, previously unannotated small proteins with important biological functions. The quality of the data presented are overall excellent and I have only minor questions that may not require additional data to address.

1. It is clear that Peg and Wg interact, and that a major phenotypic consequence of *peg* disruption is incorrect Wg diffusion and signaling during development. However, it seems likely that Peg does not specifically interact with Wg, given the homology of Peg to HSPGs and its HSPG-like function in controlling protein diffusion. HSPGs are generally thought of as cell surface-associated proteins that display glycosaminoglycans (GAGs) for interaction with many extracellular factors (though I won't use the word “nonspecific” because I don't think it's accurate). Is it possible that Peg interacts with and affects the diffusion of multiple secreted proteins, not just Wg?

See our initial clarification about homology to HSPGs above. At this point, we cannot confirm nor discard whether Peg interacts with other proteins, but we focus on the characterisation of *pegasus*'s main phenotype. However, we have changed the wording “specific binding” for “physical interaction” throughout the paper to allow for this possibility.

2. Related to the points above, the Peg-GFP fusion protein appears to migrate at 50 kD (Fig. 2D). I would expect a molecular weight of $27 + 8 \text{ kD} = 35 \text{ kD}$.

That band is Wg, we have added loading controls with Peg-GFP, and their expected weights in new westerns.

3. Finally, it is debatable whether a molecule of 80 amino acids can be accurately described as a “peptide” as in the manuscript, especially if it may adopt a three-dimensional folded structure.

See our initial discussion on molecular homologies. It is unclear whether the Kazal-like stretch in Peg holds a structure in vivo and unbound, or whether it acquires a structure upon binding, as for example antimicrobial peptides with similar sizes and molecular architecture do. We agree that the definition of “peptide” varies from smaller than 50 or 100aa. We adhere to < 100aa, because we have shown that the products encoded by small ORFs < 100aa have molecular characteristics that distinguish them from canonical, larger proteins (Couso and Patraquim 2017, Nat. Rev. Mol. Cell Biol.).

These are minor details given the clear phenotype, but important at the molecular level.

Reviewer #3 (Remarks to the Author):

The authors show that the peptide Pegasus (peg) enhances Wingless/Wnt1 protein short-range diffusion and signalling during *Drosophila* wing development. They report creating peg' null mutants and conducting experiments to compute effective diffusion coefficients and decay lengths for Wg in FRAP experiments, where the spread of Wg was measured after photo-bleaching. I was asked to evaluate specifically the authors use of a reaction diffusion model for Wnt/Wingless morphogen gradients. The authors use the well-established model of Kicheva (2007), which is properly cited throughout the manuscript. Methodologically, this is very appropriate. (I do have some fairly minor comments, see below.) Overall, I found the manuscript to be well written, scientifically sound and its result novel and interesting. Detailed comments:

- (i) page 25: "C0 is the amplitude (intensity) of the gradient at the target tissue right next to the source" This is not quite correct. In (2), C0 is the concentration of the morphogen itself (measured via the GFP intensity in the experiment), not of the gradient. In fact, an important aspect of Kichova et al.'s method is that the morphogen flux (=the magnitude of the gradient of C0) at the source $x = 0$ is constant and given by $C0/\lambda$. The authors should correct this sentence.
We have changed this sentence accordingly.
- (ii) (ii) p. 27 "[W]e ... fit spatial concentration profiles of endogenous Wg to Eq. 2": How was the fitting done? How were the reported error bounds determined? And how was the goodness of fit evaluated?
We have now clarified in methods that the fits and errors were calculated optimizing parameter search using NonlinearModelFit function in software Mathematica.
- (iii) I found it interesting that while the Wg degradation rates of the wild type and peg' are essentially the same, both the immobile fractions and the bleaching depth (=reduction in ligand concentration after bleaching) are quite distinct. Are their differences statistically significant? (The confidence intervals overlap, but that doesn't mean they're not.) And what may be the reason for this differences?
Although these differences are not statistically significant (as assessed by one tailed t -test, $p = 0.1186$ and 0.1176) they probably reflect the lower initial intensity values for peg mutants compared to wt in the measured ROI (see Fig 4 g-h, where a similar ROI as that used for the FRAP experiments is shown in context). Since the same bleaching parameters were used for the two conditions, this could give rise to a slightly increased bleaching depth in mutants compared to wild type, and consequently a decrease of the mutant immobile fraction, this can be understood from the analysis of more detailed transport models, as explained in Aguilar-Hidalgo et al. (2019 arXiv: 1909.13280).
- (iv) p. 26 The "bleaching depth" should be defined to make the manuscript more self-contained. (One may think first that this is a length, but of course it's not.)
We have now defined it as: b is the bleaching depth, that is the reduction in ligand concentration after bleaching.
- (v) p. 25, eq. (1) I found the inclusion of the production rate v slightly misleading since it's never given explicitly and in fact ultimately, the only way that the production of morphogens enters the model is via the assumed flux at the source boundary. (Kicheva et al. use a Dirac delta distribution $\delta(x)$.)
We think that it is illustrative to show the simple diffusion equation (1), and necessary to understand our methods. Furthermore, this term becomes

less relevant since we show experimentally that Wg production is not affected by Peg, as neither the Wg mRNA (by qRT-PCR) nor Wg protein (by q. Western Blot) levels are affected by Peg gain or loss of function.

Also (slightly nit-picky) I found the different sizes of the "C" in eq. (1) to be very confusing. It looks like the "C"s in $\partial C / \partial t$ and $\partial^2 C / \partial x^2$ are different than the one in $-KC$.

Fractions in inline equations appear with a reduced font size in Word.

Hopefully, since now the Kicheva et al. 2007 Science paper is cited in this section, there will be no confusions..

- (vi) p. 26 eq. (4): The explicit solution is taken from Kichova et al. (2007) and since it's not all obvious, it would be good to indicate this in the sentence directly preceding it.

We have now added this citation.

- (vi) (vii) p. 26 eq. (5) Giving the explicit form of $f(t)$ is not necessary and just bloats the manuscript.

Even though it may seem bulky we would prefer to leave the equation since it is in supplementary material and we refer to it later in methods (equation (5)).

Reviewer #4 (Remarks to the Author):

In the article "Pegasus, a small extracellular peptide enhancing the short-range diffusion of Wingless" Couso and co-workers claim that Pegasus is a short 80 aa peptide that extends the signalling range of Wg in the *Drosophila* wing imaginal disc. The authors show that Peg can also regulate signalling of Wg by analysing target gene expression and wing morphology.

The experiments are well done, and they provide convincing evidence for a role for an association between Pegasus and Wg that is important for *Drosophila* wing development. The authors have been thorough in demonstrating that Wg and Peg co-localise, co-IP, and that Wg can enhance the distance that Wg diffuses from the cell. For the most part, the data clearly represent what is asserted in the results text, except for figure 1H, which I had to zoom in extensively to detect a reduction in secreted Wg.

We have modified the figure to emphasize changes, switching the Wg-only panel to black and white.

I also think I would like more of an understanding from this paper about potential interactions between Peg and HSPG. The authors show that Pegasus is related structurally to HSPGs, which have been shown recently to have a role in promoting secreted Wnt signalling. It has been shown that glypicans are structurally similar to the CRD domain of Fzd receptors and thus allow long-range propagation of Wg. A similar analysis of Peg in its relation to the palmitoylation site of Wg would be crucial. It would be further interesting to show if overexpression of the two *Drosophila* glypicans can substitute for the loss of Peg.

See our initial comment on HSPG homologies. We have tried a similar experiment with an interaction with UAS-sugarless and we observe no correction of the phenotype. We have also included results showing no genetic nor physical interaction between Peg and the Dally-like HSPG (Fig S7).

Furthermore, given that the authors could not find evidence of a mammalian orthologue, their assertion that this is clinically relevant is somewhat unfounded.

We now present new examples of vertebrate and chordate homologues (Fig. 1 and S1), and confirmation of functional homology with fish orthologues by colocalisation of Wg and zebrafish zPeg, which also rescues Peg phenotypes. At this point we cannot confirm nor exclude the existence of mammalian orthologues. However, even in the case that these may not exist, the existence and functional homology of fish zPeg opens the door for a clinical use of Peg peptides.

The authors claim that Peg could function comparably to Spink in vertebrates. However, overexpression of Spink5 has been shown to inhibit rather than promote β -catenin Wnt signalling in oesophageal cancer (Wang, 2019), and mutations in Spink1 are associated with pancreatic cancer and a reduction in canonical Wnt signalling (Shimosegawa, 2009).

See our initial comment on HSPG homologies. Superficially, Peg resembles Spinks, but belongs to a different peptide family, and Peg does not affect Wg degradation.

Finally, it has been shown that tethered Wg can rescue the Wg mutant phenotype to a great extent - especially concerning the phenotype of the wing. Tethered Wg leads only to an overall reduction of 11% of the wing size without an apparent patterning phenotype. The authors could analyse the interaction of tethered Wg with Peg and need to discuss their findings in the light of this publication.

This was already presented and discussed in the previous version of the manuscript. We have now expanded the epistasis experiments involving WgNRT, by looking at sens expression as well as wing margin phenotypes. As explained in our

developmental model (Fig.S8), the “wing growth promoting” function of Wg happens during early third instar (72-96h AEL) and acts on Dll expression, while the main (or only) function of Wg at the presumptive wing margin after 96h AEL is to promote wing margin development, as shown by two different techniques that selectively eliminate Wg function at this stage (Couso 1994, Development; Piddini 2009 Cell).

Specific points:

It is unclear to me if Peg could act as a permissive factor allowing Wg signal transduction by activation of the formation/signalling of Wnt ligand-receptor complex. The authors may show if Lrp6 signalosome formation is altered at the plasma membrane of Peg $-/-$ cells.

Our results show an effect of Peg on Arm, but these and other experiments reveal this to be an indirect consequence of Peg effect on Wg diffusion. We showed that Peg loss of function does not enhance the Wg-NRT phenotype, strongly suggesting that Peg has no effect on signal transduction. Furthermore, we now show co-localisation of Wg and Peg in the extracellular space, i.e. before signalosome formation.

The impact of this paper could be further improved by determining if Wg can bind to other Drosophila Wnt ligands, in particular for example Wnt4, which functions indirectly through PTK7 in opposition to canonical Wnt to regulate polarity, cell migration and invasion in drosophila (Peradziryi, 2011). This could explain the role of the related Spink proteins in mammals. If OE of Drosophila peg can also function to inhibit (as well as enhance) canonical Wnt signalling in particular contexts, this may shed light on the mechanisms of Spink proteins in general and cancer in particular.

Peg does not produce any of these phenotypes, and exerts a positive influence on Wg, not negative as Wnt4. As mentioned in our initial comment, our results discard that Peg is a Spink, but rather indicate they it belongs to a new family promoting Wg diffusion.

In-situ hybridisation cannot be used as a tool to quantify mRNA expression but determine localisation. (Fig. S3AB)

In situs can detect more extensive expression. We observed normal expression with in situs and then used qRT-PCR to quantify it.

Reviewer #5 (Remarks to the Author):

The authors identified a novel small Open Reading Frame (smORF) peptide named Pegasus (Peg) that apparently augments the short-range diffusion of Wingless (Wg) in Drosophila melanogaster. Loss of Peg phenocopies a mild Wg loss of function mutant: exhibiting a reduction in chemosensory bristles in the adult wing and narrowed proneural gene expression (senseless) in the larval wing disc. Wg diffusion seems to be impeded in the large peg $-/-$ clones and in Peg mutant discs, suggesting that Peg positively promotes Wg distribution. Over-expression studies support this theory, suggesting that UAS-peg-GFP binds directly to Wg and increases Wg dispersion. The authors propose that Peg is novel contributor to Wg signaling in the fly wing.

Major Concerns

The presented data are not sufficiently rigorous to support the conclusions made.

The Peg CRISPR mutant was not rigorously tested and only one mutant allele was used.

RNAi lines and other mutant alleles that KO or KD Peg should have been used to corroborate the mutant Peg phenotype. To test if *peg*^{-/-} is a true null, in situ hybridization experiments showing the lack of *peg* mRNA expression in a *peg*^{-/-} background should have been done like they were done in wild type discs (Figure 1C-G).

We infer that the reviewer is concerned that the observed phenotypes could be due to secondary mutations in the *peg*^{Del1} chromosome. We now use a second, independently generated null allele, i.e. *PegGal4*, which replaces the entire ORF of *peg* with Gal4 (Fig 1 and S9). These mutants are both nulls as the Peg ORF is totally disrupted (Del1) or lost (PegGal4). They still produce RNA, either non-coding (Del1) or encoding Gal4 (PegGal4), but since *peg* is a coding gene with a single coding ORF, they both produce a total loss of *peg* function, which is rescued by *UAS-peg* minigene. We have another allele, hypomorphic, produced by a transposon insertion in the 5'UTR next to the ORF, thus presumably either truncating the RNA, or interfering with ORF translation, and whose similar phenotype of chemoreceptor loss is reverted by jumping out the transposon. We have not included these data because it is a hypomorphic allele whose deleterious effect on Peg peptide production cannot be quantified precisely, while we already have two alleles generated by CRISPR and mapped to the nucleotide. RNAi lines are known to produce artefacts and hypomorphic effects, so they would be inferior tools compared to our null alleles.

To study the over-expression of Peg, the authors used UAS-*peg*-GFP. Over-expression studies should have also been performed with UAS-*peg* (absence of tag) to ensure that tagging did not affect the function.

We already provided this in the previous version, see Fig 3-i uses untagged *peg*.

Related to this, if Peg promotes Wg diffusion then overexpression would be expected to lead to a local reduction of Wg levels. The data in Fig S3 suggests that UAS-*peg*-GFP only leads to a mild overexpression, compared to endogenous levels.

Reduction of Wg levels by over-diffusion is only a possibility, and would depend on Peg being able to mobilise a fraction of Wg significantly above normal production and release from the expressing cells. This, if at all possible, would rely on Peg overexpression well above physiological levels. We have not observed this effect with any of the Gal4 lines employed, and since we are aiming to compare all our observations with Peg effect on Wg *in vivo*, we are in any case aiming for near-physiological conditions in all our experiments.

Co-localization and expression studies of Peg were mostly done with the UAS/Gal4 system. This is generally problematic because both Wg and Peg and were over-expressed with the same driver and the apparent co-localization could be merely co-incidental. The two secreted proteins found in same endosomes, for example. It is certainly not proof of interaction.

This artefact would not explain the loss of function genetic interactions between Peg and Wg, but it is a fair concern when it comes to proving co-localisation. We have (laboriously) generated a *peg-Gal4* line by CRISPR-mediated gene-editing (Fig. S9) to repeat these and other observations with endogenous expression patterns (see new Fig. 3).

Critically, Tween (0.1%) should not have been used during the antibody staining because this permeabilizes the membrane and it is unclear if they are really analyzing only secreted, extracellular Wg/Peg. The standard method in the field is described in Strigini and Cohen (2000).

We have amended the method section that may have induced confusion. The FRAP experiments were carried out in vivo with GFP-tagged proteins, without either fixation, antibody staining, or Tween. These and other experiments reveal effects on Wg on a range of 3 to 4 cells (i.e. Fig. S4G, 3 μm = 1 cell diameter). This is also the functional range of Wg revealed by genetic loss of function experiments (Couso 1994,; Piddini 2009), and marked by the tsm sensilla in our images (Phillips et al. 1993, Development). Thus our well-established antibody staining technique for membrane and secreted proteins (Couso et al. 1994,; Bishop et al. 1999, Development, Galindo et al. 2002, Science, Magny et al 2013, Science, Pueyo et al. 2017 PLoS Biology, etc...), which only use one mild detergent wash after fixation with PFA and methanol, highlights the functional pool of Wg at this particular stage and in this particular function.

The IP experiment in Figure 2D is missing critical controls, so it is not certain whether Wg/Peg interact with one another. Given the pivotal nature of the proposed interaction more work should be done to validate this. For example, the levels of the pulled down GFP proteins are not shown. This is essential. It would also be more convincing if the reverse experiment was done and Peg was pulled down with an anti-Wg antibody. If this is problematic, then Wg-GFP could be used and an alternatively tagged Peg. Also it is not clear if extracellular Peg and Wg interact, the pulldown was done with whole larvae lysate. This would be easy to test in cell lines. The phenotype suggests the Wg, Peg interaction is specific, it would be more convincing to test binding to another secreted protein (Grk, Dpp, Hh).

We have not been able to carry out the reverse pull-down as we could not obtain a working anti-Peg antibody, nor a working alternative in-vivo Peg tag (note that all “standard” molecular techniques are more challenging with small ORF peptides). However, we have repeated the pulldown using Peg-Gal4 and wing discs extracts only, and included extra controls such as the GFP load. Thus the interaction observed cannot be explained by an artefactual coincidence of Peg and Wg in the expressing cells, nor by an interaction in another larval organ. The simplest explanation is that the pull down reflects the co-localisation revealed by imaging data, which also indicates that at least some co-localisation is extracellular (Fig. 2). Because we observe an effect of Peg on Wg diffusion, again the simplest explanation is that the pull-down reflects an extracellular interaction. Note that we have changed the wording across the paper from “specific binding” to “physical interaction” since we cannot discern whether Peg and Wg interact directly, or as part of the same complex. Crucially, we believe that this level of detail is not needed now to show that the primary function of Peg is to modulate Wg diffusion, which is the aim of this work, the initial characterisation of a small ORF peptide.

The authors suggest that Peg promotes Wg diffusion because it has an FS-like domain which are present in heparin-sulphate proteoglycans (HSPGs) that bind to the extracellular matrix and facilitate diffusion. Related to the above comments on – other extracellular factors that interact with HSPGS should be tested.

See our initial comment on this issue. In summary, the molecular homologies indicate that Peg is a new family of peptides in the Kazal superfamily, so a primary interaction with HSPGs and their interactors is not to be expected. Further, our new data (Fig. S7) indicates that Peg acts formally downstream of HSPGs.

The conservation analysis is weak (Figure S1). The only conclusion that is convincing is that Peg probably has a Kazal/Follistatin-like domain. A better alignment is needed. For the work

to have a significant impact on the Wnt field conservation outside of *Drosophila* would be important.

We have included more extensive homology data (Fig. 1, S1 and sup. File 1) including quantification of similarities, suggesting that Peg peptides are conserved in basal insects, basal chordates, and vertebrates (see also rescue of phenotype and co-localization with zebrafish Peg).

Lastly, the authors posit that only short range signalling is affected, yet the expression of long range (lower threshold) targets such as Dll are not tested. They should be.

We now show that Dll is unchanged in large clones (Fig. S2k).

The concept of secreted proteins modulating Wg/Wnt diffusion is not novel.

Other studies have identified similar factor which affect Wg diffusion. For example, the Nusse lab have identified Swim, a Wg binding protein, that facilitates long-range diffusion of Wg by maintaining the activity and solubility of Wg (Mulligan et al., 2012). This paper was not cited. More crucially the authors should test how Swim and Peg functionally interact. The mild phenotype of the Peg mutant suggests it is not a key factor, perhaps it is affected by Swim. For a journal with the scope and impact of Nature Communications, such considerations are de rigueur.

Although both Peg and Swim act as regulators of Wg diffusion, they differ in that Swim has been proposed to enhance long-range Wg signalling, with Swim LOF affecting Dll exclusively and having no effect on Sens expression at later third instar (Mulligan et al., 2012). Peg on the other hand appears to be required for short-range signalling, affecting Sens, but not Dll (see point above). We thus see little point in testing this interaction, but have nonetheless included this point and the cited reference at the end of our discussion. The mild phenotype of Peg is to be expected, since a diffusion-null Wg allele (WgNRT) produces a similar phenotype (Fig. 1), since cells in direct contact with Wg-expressing cells still receive Wg signal. Because the pseudo-columnar nature of the wing disc epithelium, WgNRT produces signalling in a range of 1-2 cells (Fig. 5).

Minor concerns

1) Figures are poorly presented

a. Figure 2D poor image quality

b. Figure 3 F,H graphs have different scales; labeling; fonts/typos

c. Figure S3 B', stretched graphs UAS-CG17278, (Is this UAS-peg-GFP) or is it UAS-peg?

We have addressed these points. In doing so, and because of the requests for data already included in the manuscript, we have re-arranged the figures and the results section, hopefully leading to an easier to follow manuscript.

2) Methods are limited

a. The full genetic background of each genotype is not written out which prevents a critical judgement of the experiments.

Full genotypes are now presented in methods.

b. Authors should show senseless expression in Da-Gal4 UAS-GFP and 69B-Gal4 UAS-GFP flies because these Gal4 drivers are used interchangeably and they may independently act on sens expression. Da is known to affect Sens.

We now use endogenous peg-Gal4 for rescue, and provide chemosensory bristle numbers of controls.

c. Wg-IL and Wg-NRT have similar defects but peg alterations were only tested in the Wg-NRT background, for consistency this should have been repeated with Wg-IL (Figure S1). The number of chemosensory bristles should also be shown.

We have quantified the bristles in WgIL, and as expected is similar to WgNRT (Fig. 1C). However, although WgIL affects diffusion, it has to be used at the semi-permissive temperature of 17,5°C, otherwise the Wg protein remains inside the expressing cells and is not even presented to neighbouring cells (and even so this condition is poorly viable; Gonzalez et al. 1991, MoD; Couso et al. 1994).

Unfortunately, the exact molecular effect on Wg secretion can't be quantified easily and is temperature-dependant, so this is why we have used instead WgNRT. Note that there exist other Wg alleles affecting secretion and signalling in different ways, but we do not anticipate any significant gain that would be obtained by repeating experiments with these challenging mutant conditions. We use here WgIL only to illustrate that the previously unreported phenotype of WgNRT is to be expected from a bona-fide loss of Wg secretion, and hence that experiments can be carried out using this precisely engineered allele.

3) Conflicting/unexplained phenotypes

a. It isn't explained why is there a complete loss of sens in large clones, but only narrowed presence in mutant discs (Figure 1H,I). Indeed given the expression pattern of Peg in the pouch it is not clear why lost and not reduced. It is highly expressed in both the dorsal and ventral compartments and based on the experiments in Figure 2 be able to reach the clone where Wg is present.

A quantification now shows that the average loss of sens in clones is of about 40%, although it is variable from clone to clone. The "stronger than null" phenotype observed in some clones is a classical observation and can be explained by competition from neighbouring wild-type cells for a limited resource (for example, Minute clones are out-competed for access to Dpp; Moreno et al. 2002 Nature). In our case, Peg- cells in clones must have access to less Wg than in whole Peg mutant discs, because the wt cells neighbouring the Peg clones are better at diffusing and using up the common pool of Wg.

b. sens expression is reduced in the control en>GFP, but this was not addressed by the authors. Thus the increased of sens expression with en>UAS-peg-GFP is questionable. The levels of sens are lower in the wt posterior compartment. This is why these observations are quantified comparing posterior compartments, while the anterior compartment levels are a mere internal control showing that the effect is specifically limited to the posterior compartment where enGal4 is active. This has been presented more clearly in bar charts, moving the anterior compartment data to Fig. S2.

REVIEWERS' COMMENTS

Reviewer #2 (Remarks to the Author):

While we are all sympathetic that experiments for review are often "laborious", the clarifications and additional data presented in the revised manuscript significantly improve it. While the interaction between Peg and Wg remains poorly characterized, the correction to use of "physical interaction" ("association" might be even more accurate) is an acceptable way to address this limitation. Particularly impressive is the inclusion of the zebrafish homolog complementation. Overall, this work significantly expands the smORF field.

Reviewer #3 (Remarks to the Author):

The authors have adequately addressed all my concerns.

Reviewer #4 (Remarks to the Author):

The authors addressed most of my concerns.

The assumption that Peg could also function in mobilising Wg in vertebrates has been addressed, which strengthens the core statement of the ms.

The image quality has been improved.

Fig. S7b is an excellent way to illustrate the function of Peg. However, the drawing would benefit if Peg itself would be displayed. Even if it would be partially speculative, Peg and its interaction with Wg could be shown in this cartoon and put forward a novel view for the readers.

Reviewer #5 (Remarks to the Author):

The authors performed many of the control experiments that were requested, and the manuscript is now sufficient for acceptance in Nature Communications.

-A critical objection to this study was the use of only one peg mutant that was not well-validated. This has been remedied by the development of a second null allele. Both nulls were rescued by the over expression of UAS-peg. This is now acceptable.

-The authors altered the text to emphasise that there is a physical interaction between Peg and Wg, but it is not necessarily direct (which they were unable to show with their previous and current data). This is now an acceptable conclusion and the focus of the paper relies on the function Peg aiding the short range diffusion of Wg but not the specific mechanism.

-The quality of the figures and writing in the manuscript have greatly improved, detailing a much more interesting and informative story about Wg diffusion.